# Evaluating Diversity of LLM-generated Datasets: A Classification Perspective

## Abstract

LLM-generated datasets have been recently leveraged as training data to mitigate data scarcity in specific domains. However, these LLM-generated datasets exhibit limitations on training models due to a lack of diversity, which underscores the need for effective diversity evaluation. Despite the growing demand, the diversity evaluation of LLM-generated datasets remains under-explored. To this end, we propose a diversity evaluation method for LLM-generated datasets from a classification perspective, namely, DCScore. Specifically, DCScore treats the diversity evaluation as a sample classification task, considering mutual relationships among samples. We further provide theoretical verification of the diversity-related axioms satisfied by DCScore, demonstrating it as a principled diversity evaluation method. Additionally, we show that existing methods can be incorporated into our proposed method in a unified manner. Meanwhile, DCScore enjoys much lower computational costs compared to existing methods. Finally, we conduct experiments on LLM-generated datasets to validate the effectiveness of DCScore. The experimental results indicate that DCScore correlates better with various diversity pseudo-truths of evaluated datasets, thereby verifying its effectiveness. Our code is available via: https://anonymous.4open.science/r/DCScore-ICLR/.

## 1 Introduction

Large language models (LLMs) have shown exceptional performance across a range of fields, such as chatbots (Achiam et al., 2023), computer programming (Gu, 2023), and reasoning (Yuan et al., 2024). Inspired by their remarkable capacities, some research (Ye et al., 2022; Abdullin et al., 2024; Ding et al., 2024) employs LLMs as dataset generators to mitigate the shortage of training data. Although generated data facilitates model optimization, recent studies (Yu et al., 2024; Lee et al., 2023) suggest that a lack of diversity within the dataset—measured by the variation between samples (Long et al., 2024)—may lead to performance degradation in some scenarios. To ensure a highly diverse LLM-generated dataset, previous studies have focused on improving the diversity of LLM-generated datasets through prompt design (Yu et al., 2024) and multiple topics (Wang et al., 2022). However, these studies neglect a critical aspect, i.e., evaluating the diversity of LLM-generated datasets in a reasonable manner. A principled diversity evaluation metric can provide feedback to LLM generators and guide them to generate data with higher diversity. Beyond evaluating the diversity of LLM-generated datasets, these methods can also be used for data selection (Cao et al., 2023), quantifying augmentation performance (Yang et al., 2024b; Gontijo-Lopes et al., 2020), and assessing mode collapse (Dan Friedman & Dieng, 2023). In summary, a diversity evaluation method for LLM-generated datasets is becoming increasingly crucial.

Since the diversity evaluation of LLM-generated datasets remains under-explored, a natural solution is to directly employ diversity evaluation methods from the fields of natural language processing (NLP) (Khurana et al., 2023) and machine learning (ML) (Jordan & Mitchell, 2015). Specifically, efforts to measure diversity within these domains can be summarized into three categories: *N-gram-based method* (Zhu et al., 2018; Mishra et al., 2020), *Reference-based method* (Heusel et al., 2017; Cífka et al., 2018), and *Transformation-based method* (Du & Black, 2019; Zhang et al., 2024). The n-gram-based method evaluates diversity through n-gram statistics, e.g., the distinct n-grams metric (Li et al., 2015) calculates the proportion of unique n-grams out of the total number of n-grams. This approach primarily focuses on the form difference of evaluated texts, often overlooking

semantic aspects and offering limited flexibility for evaluators. To align the diversity criteria with human judgment, the reference-based method has emerged as a promising alternative. This approach employs a reference distribution or data as an approximation of human judgment and calculates the similarity between the evaluated data and the reference data (Holtzman et al., 2019). However, the process of collecting reference data can be both time-consuming and potentially bias-inducing.

Drawing inspiration from deep representation learning (Butepage et al., 2017; Zhang et al., 2021), the transformation-based method evaluates diversity by first mapping the data into the representation space and then performing diversity summarization (Tevet & Berant, 2020). This summarization often employs high-complexity algorithms, such as eigenvalue computation. Here, various embedding functions can be employed to facilitate the transformation. A popular way is to leverage a sentence transformer, e.g., Sentence-Bert (Reimers, 2019) and SimCSE (Gao et al., 2021), to generate representations of the evaluated data and subsequently calculate the average similarity between these representations. Owing to the versatility of embedding functions, transformation-based methods can simultaneously consider extensive aspects, such as semantics, form, and style, to encode data representations, thereby providing a more comprehensive evaluation of diversity. However, this type of method struggles to offer a holistic evaluation of a dataset due to its reliance on complex diversity summarization.

In a nutshell, existing diversity evaluation methods in NLP and ML suffer from inherent limitations in the LLM-generated dataset evaluation scenario. To effectively evaluate the diversity of LLM-generated datasets, the following challenges must be tackled: (1) *Holistic Analysis*. The diversity evaluation of LLM-generated datasets is a holistic analysis task, necessitating consideration of the impact of each sample on the final evaluation results. If a new sample is added, the diversity evaluation of the newly augmented dataset should completely account for the relationship between each original sample and the new one. (2) *Axiomatic Requirements*. Prior research has suggested several axioms that diversity metrics should ideally satisfy. To align with these intuitive principles, a well-established diversity evaluation should exhibit properties corresponding to these axioms.

To sum up, the essence of diversity is associated with the identification of differences between samples, and the ability to distinguish these differences is a key element in the classification process (Quine, 1969). Motivated by this, we propose an LLM-generated dataset diversity evaluation method from a classification perspective, namely, `DCScore`. On the one hand, `DCScore` treats the evaluation of each sample in the LLM-generated dataset as a distinct classification task, effectively addressing the need for a holistic analysis. On the other hand, we provide theoretical verification that `DCScore` satisfies four axioms outlined in Leinster & Cobbold (2012), including effective number, identical samples, symmetry, and monotonicity. Meanwhile, we also observe that existing methods can be reformulated from the view of `DCScore` by leveraging different embedding and classification functions.

Our contributions can be summarized as follows:

- We propose `DCScore`, a classification-based diversity evaluation method for LLM-generated datasets. The core idea behind `DCScore` is to treat diversity evaluation as a sample classification task, enabling the capture of mutual relationships among samples.

- We theoretically validate that `DCScore` adheres to several intuitive axioms suggested by Leinster & Cobbold (2012), evidencing the superiority of `DCScore`. Additionally, we integrate existing methods into the `DCScore` framework in a unified manner.

- Extensive experiments show that `DCScore` exhibits a stronger correlation with diversity parameters and human judgment compared to baseline metrics. We also perform a computational cost experiment to confirm the lower computational expense of `DCScore`.

## 2 RELATED WORK

We give a brief literature review of diversity evaluation methods. Moreover, limited by the space, further related works on LLM dataset generators can be found in Appendix A.

## 2.1 Diversity Evaluation Methods

**N-gram-based Methods.** With the development of LLMs as dataset generators, the diversity evaluation of LLM-generated datasets has become a challenging task and remains under-explored in recent evaluation studies (Liang et al., 2022; tatsu lab, 2023). The most comparable diversity evaluation research can be traced back to studies in NLP and ML, which can be summarized into the n-gram-based method (Mishra et al., 2020), reference-based method (Heusel et al., 2017), and transformation-based method (Lai et al., 2020). The n-gram-based method is the most popular lexical diversity evaluation method, leveraging n-grams to capture differences in sentence form (Yu et al., 2024). Commonly used n-gram-based diversity metrics include distinct n-grams (*distinct-n*) (Song et al., 2024), self-BLEU (Shu et al., 2019), and ROUGE-L (Wang et al., 2022; Padmakumar & He, 2023). However, this method has limitations, as it overlooks differences in other aspects such as semantics and style.

**Reference-based Methods.** Diversity evaluation is inherently subjective, leading to a reliance on human judgment. Consequently, the reference-based method evaluates diversity by comparing the distribution of the evaluated data to that of a reference dataset (Heusel et al., 2017). MAUVE (Pillutla et al., 2021) exemplifies this idea by employing a divergence-based metric to capture correlations with human judgment. Regarding the natural language inference (NLI) training set as the reference dataset, (Stasaski & Hearst, 2022) first trains an NLI model to infer the relationship between pairs of generated texts and then calculates diversity based on these inference results. Due to the challenges in collecting reference datasets, a recent study (Le Bronnec et al., 2024) proposes evaluating diversity through precision and recall. Despite these advancements, the reference-based method remains significantly constrained because of the need for reference datasets.

**Transformation-based Methods.** The transformation-based (Lee et al., 2023) method leverages well-designed models to generate representations of the evaluated data. Then, the diversity of these representations is summarized using techniques such as eigenvalue computation (Dan Friedman & Dieng, 2023) and clustering (Du & Black, 2019). Owing to the superior performance of representation learning, this method considers various aspects of the evaluated data, including semantics, form, and style, offering greater flexibility compared to the other two methods. However, its dependence on high-complexity summarization techniques, such as eigenvalue computation, limits its scalability for comprehensive evaluation tasks, such as evaluating the diversity of LLM-generated datasets.

In summary, existing methods primarily focus on NLP and ML fields and are challenging to apply directly to overall dataset diversity evaluation. Different from the above-mentioned studies, our work is dedicated to the holistic diversity evaluation of LLM-generated datasets. Additionally, to ensure flexible evaluation, our work aims to evaluate diversity-sensitive components that impact the performance of trained models in terms of diversity.

## 3 Prelinimaries

### 3.1 LLMs as a Dataset Generator

Since the exceptional performance of LLMs, previous works (Dai et al., 2023; Yoo et al., 2021) employ LLMs as a dataset generator or for data augmentation purposes. LLMs significantly reduce the cost of label annotation and data collection (Tan et al., 2024), and in several tasks, even outperform human annotators (Gilardi et al., 2023). While some studies attempt to use LLMs to generate datasets from scratch, it is a challenging task for LLMs. In most cases, a pre-trained LLM, denoted as $\mathcal{M}$, takes the data $\mathcal{D}_{sup}$ to be augmented and the generation task $T$ as input, and outputs the augmented dataset $\mathcal{D}$. Formally, this process can be formulated as follows:

$$\mathcal{D} \leftarrow \mathcal{M}(T, \mathcal{D}_{sup}) \tag{1}$$

where $T$ can be texts describing the generation task, such as annotation. $\mathcal{D}_{sup}$, which comprises a small number of seed samples or unlabeled inputs, serves as supplementary materials to facilitate data augmentation. For example, we want LLMs to perform an annotation task for sentiment labeling, such as determining whether the sentiment is positive or negative. If we assume $\mathcal{D}_{sup}$ to be "*It's a boring movie.*", the description of $T$ could be "*The sentiment of the movie review is*". $\mathcal{D} = \{\mathcal{T}_i\}_{i=1}^{n} = \{(x_i, y_i)\}_{i=1}^{n}$ is the generated dataset with $n$ samples, where $\mathcal{T}_i = (x_i, y_i), x_i$, and $y_i$ are the input-output sample, the input text, and the output text, respectively. Let $T_{\mathcal{D}}$ denote the

Table 1: Categories of $\mathcal{D}_{sup}$. In the column of "Examples", texts belonging to $\mathcal{D}_{sup}$ are highlighted in gray , while texts associated with $T$ are marked using an underline.

| Generations | $\mathcal{D}_{sup}$ | Examples |
|---|---|---|
| $\{x_i\} \to \{y_i\}$ | $\{x_i\}$ | **Question:** It's a boring movie. The sentiment of the movie review is 
 **Answer:** Negative. |
| $\{y_i\} \to \{x_i\}$ | $\{y_i\}$ | **Question:** The movie review in positive sentiment is 
 **Answer:** Good film! |
| $\{\mathcal{T}_{seed}\} \to \{\mathcal{T}_i\}$ | $\{\mathcal{T}_{seed}\}$ | **Question:** Following are examples of movie review and their sentiment labels. Generate samples according to these examples. 
 Example 1: A boring movie. (Negative); 
 Example 2: Oh, wonderful movie! (Positive). 
 **Answer:** A meaningful movie. (Positive) |

downstream task of $\mathcal{D}$, when $T_{\mathcal{D}}$ is the question-answering task, $x_i$ and $y_i$ represent the question and answer, respectively.

It is worth noting that not all components in $\mathcal{D}$ are generated by $\mathcal{M}$, which is related to the category of $\mathcal{D}_{sup}$. As shown in Table 1, $\mathcal{D}_{sup}$ can be divided into three categories, namely input text, output text, and seed samples. In Table 1, "$\to$" and $\mathcal{T}_{seed}$ represent the direction of generation and seed samples, respectively. For example, "$x_i \to y_i$" signifies that, given input text $x_i$ denoted as $\mathcal{D}_{sup}$, $\mathcal{M}$ processes $\mathcal{D}_{sup}$ and $T$, generating the output text $y_i$.

## 3.2 PROBLEM FORMULATION

The diversity evaluation of the dataset is a sample richness evaluation problem. Based on the generation scenarios presented in Table 1, we find that in certain downstream tasks, the diversity of some components in the LLM-generated dataset does not influence the performance of the trained models. Conversely, the diversity of other components significantly impacts model performance. We refer to these components, whose diversity influences performance, as diversity-sensitive components, denoted as $\mathcal{T}_i$. For example, the input text $x_i$ in sentiment classification tasks is the diversity-sensitive component. Conversely, the output text $y_i$, which represents the sentiment label of the sample and is typically a numerical label (e.g., 0 or 1), does not influence model performance in terms of diversity. Therefore, the output text cannot be considered as the diversity-sensitive component. It should be underscored that diversity-sensitive components vary across downstream tasks. As such, the diversity evaluation of LLM-generated datasets can be transformed into the diversity evaluation of diversity-sensitive components.

Given an LLM-generated dataset $\mathcal{D} = \{\mathcal{T}_i\}_{i=1}^n$, we define $\{\tilde{\mathcal{T}}_i\}_{i=1}^n$ as a collection of diversity-sensitive components. Hence, the problem of diversity evaluation of $\mathcal{D}$ can be formulated as follows:

$$\text{DiversityScore} \leftarrow \text{Eval}(\{\tilde{\mathcal{T}}_i\}_{i=1}^n) \tag{2}$$

where Eval is the diversity evaluation function, which takes $\{\tilde{\mathcal{T}}_i\}_{i=1}^n$ as input and outputs the diversity score DiversityScore of $\mathcal{D}$.

## 4 PRESENT WORK

In this section, we first introduce our proposed method from a classification perspective, namely, DCScore. Then, we present the properties of DCScore followed by theoretical proofs. Finally, we integrate existing methods into a unified framework viewed from the perspective of DCScore.

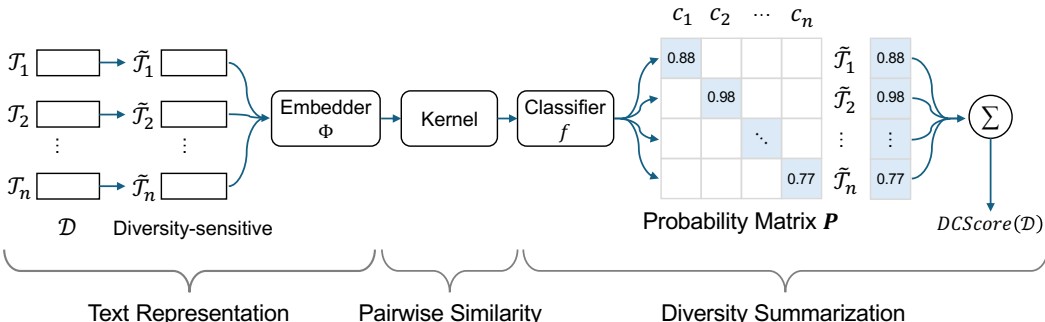

Figure 1: Illustration of the computation of DCScore.

## 4.1 DCSCORE: DIVERSITY CALCULATION FROM A CLASSIFICATION PERSPECTIVE

To tackle the holistic analysis and axiomatic requirements challenges, we propose DCScore to evaluate diversity from a classification perspective. According to the problem formulation in section 3.2, DCScore aims to evaluate LLM-generated datasets by evaluating the diversity of diversity-sensitive components. As shown in Figure 1, DCScore can treat diversity evaluation as a classification task due to the inherent sample difference measurement of diversity evaluation. The difference between samples can be measured by a $n$-classification task where the evaluation of $n$ sample datasets involves $n$ $n$-classification tasks, with each sample corresponding to a category.

Let $\mathcal{D} = \{\mathcal{T}_i\}_{i=1}^n = \{(x_i, y_i)\}_{i=1}^n$ denote an LLM-generated dataset comprising $n$ input-output samples, and $\{\tilde{\mathcal{T}}_i\}_{i=1}^n$ represents the diversity-sensitive components. DCScore adheres to the paradigm of the transformation-based method to evaluate the diversity of $\mathcal{D}$. Specifically, given $\tilde{\mathcal{T}}_i$, DCScore first applies an embedding function $\Phi$ to extract the sample representation $\mathbf{h}_i = \Phi(\tilde{\mathcal{T}}_i)$. For all samples in $\mathcal{D}$, we obtain the sample representation matrix $\mathbf{H} \in \mathbb{R}^{n \times d}$ across all samples, where $d$ denotes the dimension of sample representations. Subsequently, DCScore utilizes a kernel function Kernel to calculate a kernel matrix $\mathbf{K}$, where $\mathbf{K} \in \mathbb{R}^{n \times n}$ and entry $\mathbf{K}[i, j]$ represents similarity between $\tilde{\mathcal{T}}_i$ and $\tilde{\mathcal{T}}_j$. From a classification perspective, $\mathbf{K}[i, j]$ can be considered as the logit of $\tilde{\mathcal{T}}_i$ being classified into category $c_j$. where $c_j$ corresponds to $\tilde{\mathcal{T}}_j$. Formally, the aforementioned process can be formulated as follows:

$$\mathbf{H} = \Phi(\{\tilde{\mathcal{T}}_i\}_{i=1}^n), \mathbf{K} = \text{Kernel}(\mathbf{H}), \tag{3}$$

where Kernel$(\cdot)$ calculates pairwise similarity, with viable options including inner product and RBF kernel. For $\Phi$, a more expressive embedding function can be employed, such as one trained using a well-designed framework like Sentence-Bert (Reimers, 2019).

Based on the kernel matrix $\mathbf{K}$, DCScore leverages a classification function with $\mathbf{K}$, denoted as $f_{\mathbf{K}}$, to compute the classification probability matrix $\mathbf{P} \in \mathbb{R}^{n \times n}$. There are several potential choices for $f_{\mathbf{K}}$, among which a natural option is the Softmax function, Specifically, for $\tilde{\mathcal{T}}_i$, the probability that $\tilde{\mathcal{T}}_i$ is classified as category $c_j$ can be formulated as follows:

$$P(c = c_j | \tilde{\mathcal{T}}_i) = \mathbf{P}[i, j] = f_{\mathbf{K}}(\mathbf{K}[i, j]) = \frac{\exp(\mathbf{K}[i, j]/\tau)}{\sum_j \exp(\mathbf{K}[i, j]/\tau)}, \tag{4}$$

where $\tau$ is a temperature hyperparameter to control the classification resolution. A smaller $\tau$ amplifies sample similarity differences, implying a higher classification resolution, while a larger value yields the opposite effect.

According to equation 4, if the evaluated dataset exhibits high sample richness, indicating greater diversity, each sample is likely to be classified into its own category. Conversely, if diversity is low, all samples may be classified into a single category. Based on $\mathbf{P}$, DCScore calculates diversity of $\mathcal{D}$ as the trace of $\mathbf{P}$. In summary, the definition of diversity measurement from the classification perspective is as follows:

**Definition 1** (DCScore). *Let $\mathcal{D} = \{\mathcal{T}_i\}_{i=1}^n$ denote the LLM-generated dataset with $n$ samples, and let $\{\tilde{\mathcal{T}}_i\}_{i=1}^n$ represent a set of diversity-sensitive components within $\{\mathcal{T}_i\}_{i=1}^n$. Denote $P_i$ as the*

*classification probability vector of $\tilde{\mathcal{T}}_i$. By conducting the classification task for all $\tilde{\mathcal{T}}_i$ and obtaining the probability matrix $\boldsymbol{P} = [P_1, P_2, ..., P_n]$, DCScore for $\mathcal{D}$ is defined as the trace of $\boldsymbol{P}$:*

$$\text{DCScore}(\mathcal{D}) = \text{tr}(\boldsymbol{P}) = \sum_{i=1}^{n} \boldsymbol{P}[i, i]. \tag{5}$$

## 4.2 PROPERTIES OF DCSCORE

We provide theoretical proof that DCScore satisfies several axioms (Leinster & Cobbold, 2012) defined for a principled diversity metric. Specifically, DCScore meets four axioms: effective number, identical samples, symmetry, and monotonicity axioms. These guarantees ensure a reasonable and robust diversity evaluation. The matched axioms of our proposed method are outlined below, while their proofs can be found in Appendix B due to space constraints.

- **Effective number**: Diversity should be defined as the effective number of samples in a dataset, ranging from 1 to $n$. DCScore meets this axiom, as evidenced by its behavior: DCScore equals 1 when all samples in $\mathcal{D}$ are identical and equals $n$ when all samples are distinct.

- **Identical samples**: Given two identical datasets $\mathcal{D}_1$ and $\mathcal{D}_2$, the diversity of the synthetic dataset $\mathcal{D}'$ generated by merging these two datasets remains unchanged. The values of DCScore are the same across $\mathcal{D}_1$, $\mathcal{D}_2$, and $\mathcal{D}'$, i.e.,

$$\text{DCScore}(\mathcal{D}_1) = \text{DCScore}(\mathcal{D}_2) = \text{DCScore}(\mathcal{D}'). \tag{6}$$

- **Symmetry**: Diversity remains constant regardless of the order of the samples, exhibiting permutation invariance. Let $\pi$ denote the permutation function for the sample order, DCScore remains unchanged for any sample permutation of $\mathcal{D}$, i.e.,

$$\text{DCScore}(\mathcal{D}) = \text{DCScore}(\pi(\mathcal{D})). \tag{7}$$

- **Monotonicity**: The diversity of a dataset decreases as the similarity between its samples increases. Given two datasets $\mathcal{D}_1$ and $\mathcal{D}_2$, and a new sample $\mathcal{T}_{n+1}$, where the samples in $\mathcal{D}_1$ and $\mathcal{D}_2$ are entirely different, and $\text{DCScore}(\mathcal{D}_1) = \text{DCScore}(\mathcal{D}_2) = n$. If $\mathcal{T}_{n+1}$ is more similar to the samples in $\mathcal{D}_2$ than to those in $\mathcal{D}_1$ and is added to both datasets, then for the merged datasets $\mathcal{D}'_1$ and $\mathcal{D}'_2$, DCScore satisfies the following equation.

$$\text{DCScore}(\mathcal{D}'_1) > \text{DCScore}(\mathcal{D}'_2). \tag{8}$$

## 4.3 A UNIFIED MODELING FOR EXISTING METHODS

To further understand DCScore, we compare it with existing methods, including Distinct-n (Li et al., 2015), K-means inertia (Du & Black, 2019), and VendiScore (Dan Friedman & Dieng, 2023). We find that these methods can be integrated into the DCScore framework, which consists of three stages: *Text Representation*, *Pairwise Similarity*, and *Diversity Summarization*. In the text representation stage, sample representations are extracted, which then serve as the basis for measuring similarity between samples in the pairwise similarity stage. Finally, the diversity summarization stage evaluates the dataset's diversity by aggregating the pairwise similarities of all samples. Based on equation 3-5, DCScore is structured into these three stages as follows:

$$\begin{aligned} &\textit{Text Representation: } \mathbf{H} = \Phi(\{\tilde{\mathcal{T}}_i\}_{i=1}^{n}), \\ &\textit{Pairwise Similarity: } \mathbf{K} = \text{Kernel}(\mathbf{H}), \\ &\textit{Diversity Summarization: } \mathbf{P}[i, j] = f_{\mathbf{K}}(\mathbf{K}[i, j]), \end{aligned} \tag{9}$$

$$\text{DCScore}(\mathcal{D}) = \text{tr}(\mathbf{P}) = \sum_{i=1}^{n} \mathbf{P}[i, i].$$

In this regard, three existing methods are summarized into three stages in Table 2, with further details provided in Appendix G. *Distinct-n* calculates the proportion of unique n-grams to all n-grams within a concatenated dataset, where the concatenation, unique operation, and n-grams are

Table 2: Existing methods are modeled into the framework of `DCScore`.

| | Text Representation | Pairwise Similarity | Diversity Summarization |
|---|---|---|---|
| **Distinct-n** (Li et al., 2015) | $\text{n-grams}(\text{Concat}(\mathcal{D}))$ | $\text{Unique}(\text{n-grams}(\text{Concat}(\mathcal{D})))$ | $\frac{|\text{Unique}(\text{n-grams}(\text{Concat}(\mathcal{D})))|}{|\text{n-grams}(\text{Concat}(\mathcal{D}))|}$ |
| **K-means inertia** (Du & Black, 2019) | $\mathbf{H} = \Phi(\{\tilde{\mathcal{T}}_i\}_{i=1}^n)$ | $\mathcal{C} = \text{K-means}(\mathbf{H})$ | $\sum_{\mathbf{c}_k \in \mathcal{C}, \mathbf{h}_j \in \mathbf{H}_{\mathbf{c}_k}} (\mathbf{h}_j - \mathbf{c}_k)^2$ |
| **VendiScore** (Dan Friedman & Dieng, 2023) | $\mathbf{H} = \Phi(\{\tilde{\mathcal{T}}_i\}_{i=1}^n)$ | $\mathbf{K} = \text{Kernel}(\mathbf{H})$ | $\exp\left(-\sum_{i=1}^n \lambda_i \log \lambda_i\right)$ |

denoted as $\text{Concat}(\cdot)$, $\text{Unique}(\cdot)$, and $\text{n-grams}(\cdot)$, respectively. *K-means inertia* and *VendiScore* are transformation-based methods that perform clustering and eigenvalue computation in the representation space. For *K-means inertia*, $\text{K-means}(\cdot)$, $\mathcal{C}$, and $\mathbf{c}_k \in \mathcal{C}$ represent k-means clustering, the cluster centroid set, and the $k$-th cluster centroid. For *VendiScore*, $\lambda_i$ is the $i$-th eigenvalue of $\mathbf{K}/n$.

In summary, existing methods can be incorporated into the `DCScore` framework, indicating that `DCScore` operates as a higher-level method. `DCScore` also follows a transformation-based approach, resulting in similar processes in the text representation and pairwise similarity stages as *K-means inertia* and *VendiScore*. However, `DCScore` generally exhibits lower complexity, as detailed in Appendix F. It is worth noting that equation 9 is not the only implementation of `DCScore` from a classification perspective, and other implementation methods will be explored in future work.

## 5 EXPERIMENTS

### 5.1 EXPERIMENTAL SETTINGS

To verify the effectiveness of `DCScore`, we conduct a series of correlation experiments following the setting in Tevet & Berant (2020). As shown in Figure 2, we evaluate $l$ generated datasets to obtain $l$ diversity scores. Subsequently, we calculate the correlation between these diversity scores and the corresponding diversity pseudo-truth for each dataset. Due to the unavailability of diversity ground-truth of LLM-generated datasets, we employ the softmax temperature $\tau_g$ of dataset generation, human judgment, and LLM evaluation as the diversity pseudo-truth. For $\tau_g$, previous works (Caccia et al., 2018; Tevet & Berant, 2020; Chung et al., 2023) have demonstrated a positive correlation between $\tau_g$ and the diversity of generated texts, making $\tau_g$ as a reasonable diversity pseudo-truth. In this regard, LLMs with lower $\tau_g$ generate less

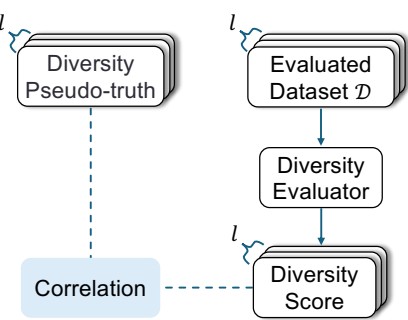

Figure 2: Experimental settings of correlation evaluation.

diverse content, whereas higher $\tau_g$ values yield more diverse content. To evaluate the correlation between calculated diversity scores and diversity pseudo-truths, we employ Spearman's $\rho$ (Spearman, 1961), a measure of rank correlation ranging from -1 to 1, with higher absolute values indicating stronger correlations. Due to space limitations, we present detailed experimental settings in Appendix D.

**Datasets.** We utilize two categories of datasets in our experiments: our generated datasets and publicly available generated datasets. Our generated datasets are generated through two data generation strategies (Li et al., 2023): *zero-shot* and *few-shot* settings. Additionally, we generate datasets for two natural language generation tasks: *text classification* and *story completion*. We utilize three publicly available existing datasets, including SST2 (Socher et al., 2013), Yelp (Zhang et al., 2015), and AG News (Zhang et al., 2015), and their AttrPrompt-augmented version from Yu et al. (2024). Detailed information about these datasets can be found in Appendix D.1.

**Generation Models.** To generate datasets through zero-shot and few-shot settings, we employ two commonly used LLMs as our dataset generators, including Llama2-13B (13B) and Llama2-70B (70B) (Touvron et al., 2023).

Table 3: Correlation (Spearman's $\rho$) between temperature and diversity evaluation methods on datasets generated by different settings (*Zero-shot* or *Few-shot*). Spearman's $\rho$ varies between -1 and +1 with 0 implying no correlation. Best results are indicated in **bold**.

| Methods | Zero-shot setting | | | | Few-shot setting | | | |
| --- | --- | --- | --- | --- | --- | --- | --- | --- |
| | Text classification | | Story completion | | Text classification | | Story completion | |
| | 13B | 70B | 13B | 70B | 13B | 70B | 13B | 70B |
| Distinct-n | 0.9909 | **0.9870** | 0.9766 | 0.9701 | 0.9857 | 0.9766 | 0.9779 | 0.9935 |
| K-means Inertia | -0.1143 | 0.9688 | 0.9454 | 0.8727 | 0.7104 | 0.7273 | 0.9662 | 0.9662 |
| VendiScore | **0.9961** | 0.9818 | **0.9870** | **0.9922** | **0.9909** | 0.9857 | **0.9857** | 0.9961 |
| DCScore | **0.9961** | 0.9779 | 0.9844 | 0.9792 | **0.9909** | **0.9883** | **0.9857** | **0.9974** |

**Baseline Methods.** We compare DCScore with three baseline methods shown in Section 4.3, i.e., *Distinct-n* (Li et al., 2015), *K-means inertia* (Du & Black, 2019), and *VendiScore* (Dan Friedman & Dieng, 2023).

## 5.2 CORRELATION EVALUATION

In this subsection, we investigate the correlation between the diversity evaluation of DCScore and diversity pseudo-truth, such as $\tau_g$ and human judgment. We compare DCScore with all baseline methods on both our generated datasets and the publicly available generated datasets.

### 5.2.1 CORRELATION WITH GENERATION TEMPERATURE $\tau_g$

**Evaluation on our generated datasets.** We evaluate the performance of DCScore on our generated datasets with varying $\tau_g$, ranging from 0.2 to 1.2 at 0.05 intervals. Limited by the space, we present more information about our generated datasets in Appendix D.1.1. Table 3 displays the correlation results of all methods. All methods accurately capture the true diversity of generated datasets, as demonstrated by high Spearman's $\rho$ values. DCScore performs on par with VendiScore while providing better scalability for larger LLM-generated datasets, as discussed in Section 5.3. Additionally, DCScore outperforms all baseline methods under the few-shot setting across all datasets, highlighting its effectiveness. *K-means Inertia* exhibits the weakest correlation on the text classification dataset generated by the 13B model under the zero-shot setting, potentially due to its sensitivity to the number of cluster centroids. Overall, DCScore outperforms all baselines in most cases, and its evaluation results exhibit a strong correlation with the diversity pseudo-truth.

**Visualization.** We further provide a visualization of the diversity evaluation results for DCScore. For each generated dataset, we prompt LLMs to produce 10 distinct answers corresponding to a single prompt, forming an evaluation batch. We then assess the diversity of these generated datasets using the batch evaluation protocol outlined in Appendix D.2.3. Ideally, a completely diverse dataset may yield a diversity score of 10.

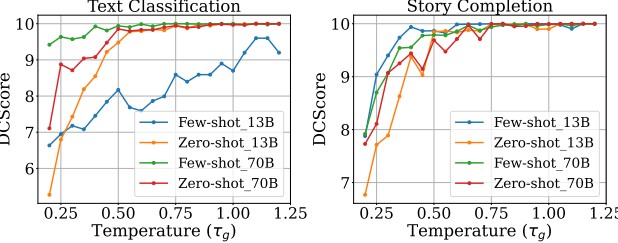

Figure 3: Diversity evaluation (DCScore) w.r.t dataset generated using different temperatures ($\tau_g$). DCScore's evaluation shows a strong correlation with $\tau_g$, indicating its effectiveness in evaluating the intrinsic diversity of the dataset.

As shown in Figure 3, DCScore exhibits a strong positive correlation with $\tau_g$, consistent with its impact on content diversity. In most cases, when $\tau_g > 0.75$, DCScore scores a generated dataset with a diversity value of approximately 10. In the text classification task, the 13B generation model under the few-shot setting demonstrates a distinct diversity change pattern compared to others. This phenomenon arises from the 13B generation model's inability to follow more complex instructions, resulting in limited diversity improvement.

Table 4: Pairwise correlation (Spearman's $\rho$) between human, temperature ($\tau_g$), and DCScore. DCScore indicates a strong correlation with human judgment.

|  | **Story-Few** | **Story-Zero** | **Text-Few** | **Text-Zero** |
|---|---|---|---|---|
| Human-DCScore | $0.9040_{\pm0.04}$ | $0.7870_{\pm0.10}$ | $0.7915_{\pm0.16}$ | $0.8798_{\pm0.10}$ |
| $\tau_g$-DCScore | $0.9086_{\pm0.07}$ | $0.7829_{\pm0.16}$ | $0.8400_{\pm0.16}$ | $0.8971_{\pm0.07}$ |
| $\tau_g$-Human | $0.9276_{\pm0.02}$ | $0.9194_{\pm0.06}$ | $0.9770_{\pm0.02}$ | $0.9255_{\pm0.08}$ |

**Evaluation on existing datasets.** Additionally, we present the diversity evaluation results on three publicly available datasets in Figure 4. By dividing each result by the maximum diversity evaluation value across three datasets, we normalize all results to the range [0,1].

We observe that, compared to baselines, the evaluation results of DCScore exhibit a similar changing trend across three datasets, highlighting its effectiveness. Additionally, both DCScore and VendiScore demonstrate a greater ability to discern diversity between different datasets compared to the other two methods.

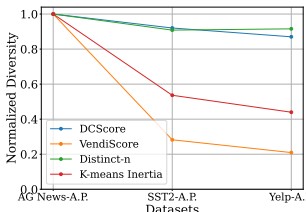

Figure 4: Diversity evaluation on existing generated datasets.

### 5.2.2 CORRELATION WITH HUMAN JUDGMENT

Diversity evaluation is a subjective task, and an ideal method should align well with human judgment. Therefore, we investigate the correlation between DCScore and human judgment. To mitigate evaluators' bias, we enlist three individuals to perform pairwise diversity comparisons among datasets with varying $\tau_g$ values and report the diversity ranking by averaging the win rate across evaluators. We conduct all evaluations five times to report average results. Moreover, we exclusively use the 70B model for dataset generation to ensure better content creation.

Table 4 presents pairwise correlation between human judgment, $\tau_g$, and DCScore. Table 4 indicates a strong correlation between human judgment and $\tau_g$, supporting the use of human judgment as a diversity pseudo-truth. Based on this observation, DCScore performs better in two settings: **Story-Few** (story completion task generation under the few-shot setting) and **Text-Zero** (text classification task generation under the zero-shot setting). This is confirmed by higher human-DCScore correlation in these two settings. In contrast, for **Story-Zero** and **Text-Few** settings, we observe more identical content in the initial portions of the diversity-sensitive components within an evaluation batch. In these cases, human evaluators tend to disregard the identical content and base their judgments on the latter sections. However, DCScore is affected by the identical content, resulting in a lower pairwise correlation. Despite this, the correlation remains strong, as demonstrated by previous studies (Akoglu, 2018).

### 5.3 COMPUTATIONAL COST

The computational cost is crucial in diversity evaluation methods, especially with the increasing sample sizes of LLM-generated datasets. For a fair comparison, we only present the computation times of transformation-based methods: DCScore, *K-means Inertia*, and *VendiScore*. We truncate the text length of three publicly available datasets to 50 tokens and record the computation times of three transformation-based methods with varying sample sizes in the range of $\{100, 500, 1000, 2000, 4000\}$.

As shown in Figure 5, we repeat the experiments five times to report the final results. DCScore and *K-means Inertia* exhibit nearly identical computation times. However, DCScore significantly outperforms *K-means Inertia* in correlation with $\tau_g$, as evidenced in Section 5.2.1. Compared to *VendiScore*, DCScore demonstrates a speed advantage of approximately 16%, or more than one second, when processing 4000 samples. Analyzing the time complexity of these two methods, and disregarding the selection of the kernel function, we find that for a dataset with $n$ samples, where $n \gg d$ is not satisfied, the computational complexity of DCScore in diversity summarization is $\mathcal{O}(n^2)$ due to the softmax computation. In contrast, *VendiScore* requires finding the eigenvalues of

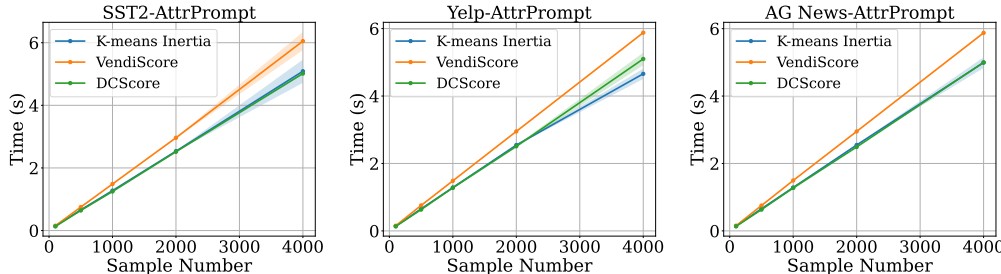

Figure 5: Computation times results under different sample sizes. `DCScore` outperforms all baseline methods in terms of computational cost.

an $n \times n$ matrix, resulting in a computational complexity of $\mathcal{O}(n^3)$. Consequently, `DCScore` offers significantly lower time complexity than *VendiScore* while sacrificing little in diversity evaluation performance. However, as detailed in the complexity analysis shown in Appendix F, when $n \gg d$ and inner products are used as the kernel function, the total complexity of *VendiScore* can be reduced to $\mathcal{O}(d^2 n)$. For a fair comparison, we evaluate computational costs on larger datasets, i.e., satisfying $n \gg d$. The detailed experimental results and analysis are presented in Appendix E.3.

## 5.4 HYPERPARAMETERS SENSITIVITY

According to equation 4, the temperature ($\tau$) in the Softmax function is a critical hyperparameter that affects classification resolution. To investigate this, we conduct a hyperparameter sensitivity analysis of `DCScore` w.r.t. $\tau$ on our generated datasets used in Section 5.2.1. We vary $\tau$ within the range of $\{0.0001, 0.001, 0.1, 0.5, 1, 10\}$. Figure 6 presents hyperparameter sensitivity results for two natural language generation tasks: text classification

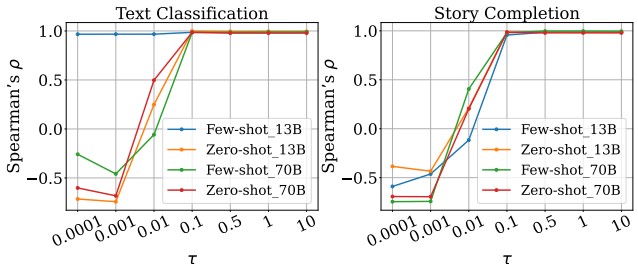

Figure 6: Hyperparameter sensitivity analysis w.r.t $\tau$ on our generated datasets.

and story completion. Overall, lower $\tau$ values result in lower Spearman's $\rho$, even indicating a negative correlation, while higher $\tau$ values do the opposite. From equation 4, a higher $\tau$ reduces pairwise similarity differences, leading to a more uniform distribution of classification probabilities for each sample. This phenomenon can be regarded as a lower classification resolution, i.e., the classification function $f_{\mathbf{K}}$ has poorer discrimination power. Furthermore, the correlation result of the 13B generation model under the few-shot setting for the text classification task remains stable despite variations in $\tau$. This phenomenon has the same explanation as in Figure 3.

## 6 CONCLUSION

In this work, we investigate the diversity evaluation of LLM-generated datasets, a topic systematically under-explored in existing research. To this end, we present `DCScore`, a diversity evaluation method from a classification perspective. `DCScore` regards the holistic diversity evaluation as the classification task at the sample level, thereby facilitating the capture of mutual relationships between samples. We provide theoretical guarantees demonstrating that `DCScore` meets the axiom requirements (Leinster & Cobbold, 2012) for a principled diversity evaluation method. Furthermore, we show that existing methods can be unified within the `DCScore` framework. Experiments on LLM-generated datasets reveal that `DCScore` exhibits better correlations with various diversity pseudo-truths, such as $\tau_g$ and human judgment. Meanwhile, `DCScore` exhibits significantly lower computational cost compared to transformation-based counterparts. Finally, we hope our work encourages future research to pay more attention to the diversity of LLM-generated datasets and promotes the wider application of these datasets.

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

APPENDICES

# A ADDITIONAL RELATED WORK

Limited by the space, we provide a literature review of the LLM dataset generator and application of diversity evaluation methods as follows.

## A.1 LLM DATASET GENERATOR

**Prompt-guided and Dataset-guided Strategies.** Recent studies (Ding et al., 2022; Chung et al., 2023) leverage LLMs to augment existing datasets or generate a dataset from scratch, demonstrating the effectiveness in improving dataset quality and reducing data collection costs. Generally, efforts to employ LLMs as dataset generators can be categorized into three strategies: *Prompt-guided* (Li et al., 2023), *Dataset-guided* (Ye et al., 2022), and *Instruct-guided* (Samuel et al., 2023). The prompt-guided strategy, a prominent data augmentation approach using LLMs, involves designing task-specific prompts to guide LLMs to augment data in a few-shot (Yoo et al., 2021) or zero-shot (Mahmoudi et al., 2024) manner. Due to its simplicity and effectiveness, subsequent works extend this strategy to various scenarios, such as medical (Yuan et al., 2023), person retrieval (Li et al., 2024b), and social media scenario (dos Santos et al., 2024). However, simple prompt engineering has limitations in fully exploiting the capabilities of LLMs, leading to the development of multi-level prompt designs (Ye et al., 2024) and targeted sample augmentation (Yang et al., 2024a). To further harness the potential of LLMs, the dataset-guided strategy employs LLMs to generate a training set and then trains a task-specific model to annotate unlabeled data (Sahu & Laradji, 2024). The dataset-guided strategy aims to approximate the distribution of targeted scenarios, but it is currently only applicable to text classification tasks.

**Instruct-guided Strategy.** Previous studies (White et al., 2023) indicate that the design of prompts significantly impacts the performance of LLMs, spurring research into the instruct-guided strategy. Generally speaking, the instruct-guided strategy leverages LLMs to generate instructions that guide another LLM in dataset generation (Evuru et al., 2024). These instructions typically relate to context (Samuel et al., 2023), criteria (Huang et al., 2023), and tasks (Wang et al., 2022). To further improve the quality of instructions, efforts have been concentrated on selecting optimal instructions (Li et al., 2024a), integrating soft instructions (Chen et al., 2023), and implementing self-correction mechanisms (Gupta et al., 2023). In a nutshell, LLMs are employed to generate or augment datasets through prompt engineering and multi-step strategies, which encompass various application scenarios and downstream tasks. Meanwhile, the diversity of LLM-generated datasets emerges as a critical factor in measuring data quality. In our work, we focus on the diversity evaluation of LLM-generated datasets derived from any dataset generation strategies.

## A.2 APPLICATION OF DIVERSITY EVALUATION METHODS

**Quantifying Augmentation Performance.** As data augmentation becomes an essential component in the training of deep neural networks (Zhang, 2017; Park et al., 2019), researchers gradually explore a better quantification of the quality of data augmentation. Some studies (Cubuk et al., 2019) suggest that the effectiveness of data augmentation arises from the increased diversity of the data. Inspired by this observation, a series of studies have introduced diversity evaluation metrics into the performance assessment of data augmentation strategies. Specifically, they consider diversity as one aspect of evaluating the quality of augmented data, thereby determining the effectiveness of data augmentation. For instance, Gontijo-Lopes et al. (2020) utilize the fundamental idea that models find it more challenging to fit more diverse data, comparing metrics such as training loss and training time before and after augmentation to assess diversity. Similarly, Yang et al. (2024b) evaluate diversity by examining the eigenvalues and eigenvectors of the similarity matrix of samples before and after augmentation.

**Evaluating Mode Collapse.** Generative adversarial networks (GANs) (Goodfellow et al., 2020) suffer from a well-known phenomenon called mode collapse, which can result in a lack of diversity in the generated samples (Dieng et al., 2019). Consequently, existing studies assess mode collapse by evaluating the diversity of the generated samples. For instance, a common approach is to train an MNIST classifier and then count the number of unique classes predicted for the generated samples. Following this paradigm, VendiScore (Dan Friedman & Dieng, 2023) compares the generation diversity of PresGAN (Dieng et al., 2019) and Self-conditioned GAN (Liu et al., 2020). Addition-

ally, some studies (Yu et al., 2017; Zhu et al., 2018; Caccia et al., 2018) employ different metrics to evaluate the diversity of generated samples from GANs.

**Other Applications.** In addition to the aforementioned applications, diversity evaluation metrics have valuable applications in various areas, including sample selection for datasets (Cao et al., 2023), enhancing robustness (Lee et al., 2022), and eliminating biases within datasets (Huber et al., 2024).

## B    PROOF OF PROPERTIES OF DCSCORE

We theoretically confirm that DCScore satisfies several intuitive axioms pointed out by previous studies (Leinster & Cobbold, 2012), thereby demonstrating its superiority.

- **Effective number (Restated)**: Diversity should be defined as the effective number of samples in a dataset, ranging from 1 to $n$. DCScore meets this axiom, as evidenced by its behavior: DCScore equals 1 when all samples in $\mathcal{D}$ are identical and equals $n$ when all samples are distinct.

    *Proof.* For DCScore, if all samples in a dataset are the same, the probability of any given sample being classified into all categories is the same, i.e., for all $i, j = \{1, 2, ..., n\}$, $\mathbf{P}[i, i] = \mathbf{P}[i, j] = \frac{1}{n}$. Then, we have DCScore $= \sum_{i=1}^{n} \frac{1}{n} = 1$. If all samples in the dataset are distinct, for all $i, j = \{1, 2, ..., n\}$, $\mathbf{P}[i, i] = 1$. In other words, the classification function confidently predicts that $\tilde{\mathcal{T}}_i$ belongs to the $i$-th category. Then, we have DCScore tending to $n$. □

- **Identical samples (Restated)**: Given two identical datasets $\mathcal{D}_1$ and $\mathcal{D}_2$, the diversity of the synthetic dataset $\mathcal{D}'$ generated by merging these two datasets remains unchanged. The values of DCScore are the same across $\mathcal{D}_1$, $\mathcal{D}_2$, and $\mathcal{D}'$, i.e.,

$$\mathrm{DCScore}(\mathcal{D}_1) = \mathrm{DCScore}(\mathcal{D}_2) = \mathrm{DCScore}(\mathcal{D}'). \tag{10}$$

    *Proof.* Assuming that $\mathcal{D}_1$ and $\mathcal{D}_2$ are completely identical, and the samples within each dataset are entirely different, i.e., $\mathrm{DCScore}(\mathcal{D}_1) = \mathrm{DCScore}(\mathcal{D}_2) = n$. Let $\mathbf{P} = [P_1, ..., P_n, ..., P_{2n}]$ denote the probability matrix of the merged dataset $\mathcal{D}' = \mathcal{D}_1 \cup \mathcal{D}_2 = \{\mathcal{T}_i\}_{i=1}^{2n}$. For $1 \leq i \leq n$, $\mathcal{T}_i = \mathcal{T}_{2i}$, where $\mathcal{T}_i \in \mathcal{D}_1$, $\mathcal{T}_{2i} \in \mathcal{D}_2$. Consequently, for each diversity-sensitive component $\tilde{\mathcal{T}}_i$ in $\mathcal{D}'$, $\mathbf{P}[i, i] = \mathbf{P}[i, 2i] = \frac{1}{2}$. Finally, $\mathrm{DCScore}(\mathcal{D}') = \sum_{i=1}^{2n} \frac{1}{2} = n$.
    However, the assumption that all samples in the dataset are completely different may be too stringent. We further provide a proof with a more relaxed assumption. Suppose that $\mathcal{D}_1$ and $\mathcal{D}_2$ are completely identical, with $\mathbf{K}_1$ and $\mathbf{K}_2$ denoting the kernel matrices for $\mathcal{D}_1$ and $\mathcal{D}_2$, respectively. In this case, we have $\mathbf{K}_1 = \mathbf{K}_2$ as follows:

$$\mathbf{K}_1 = \mathbf{K}_2 = \begin{bmatrix} k_{1,1}^1 & k_{1,2}^1 & \cdots & k_{1,n}^1 \\ k_{2,1}^1 & k_{2,2}^1 & \cdots & k_{2,n}^1 \\ \vdots & \vdots & \ddots & \vdots \\ k_{n,1}^1 & k_{n,2}^1 & \cdots & k_{n,n}^1 \end{bmatrix} = \begin{bmatrix} k_{1,1}^2 & k_{1,2}^2 & \cdots & k_{1,n}^2 \\ k_{2,1}^2 & k_{2,2}^2 & \cdots & k_{2,n}^2 \\ \vdots & \vdots & \ddots & \vdots \\ k_{n,1}^2 & k_{n,2}^2 & \cdots & k_{n,n}^2 \end{bmatrix}. \tag{11}$$

    According to equation 4, for the $i$-th diversity-sensitive component in $\mathcal{D}_1$ and $\mathcal{D}_2$, the probability of being classified as category $c_i$ can be computed as follows:

$$\mathbf{P}_1[i, i] = \mathbf{P}_2[i, i] = \frac{k_{i,i}^1}{\sum_j k_{i,j}^1} = \frac{k_{i,i}^2}{\sum_j k_{i,j}^2}. \tag{12}$$

    For a merged dataset $\mathcal{D}' = \mathcal{D}_1 \cup \mathcal{D}_2 = \{\mathcal{T}_i\}_{i=1}^{2n}$, when $1 \leq i \leq n$, we have $\mathcal{T}_i = \mathcal{T}_{2i}$, where $\mathcal{T}_i \in \mathcal{D}_1$, $\mathcal{T}_{2i} \in \mathcal{D}_2$. Since the newly added data samples do not affect the pairwise

similarity, the kernel matrix $\mathbf{K}^{'}$ for $\mathcal{D}^{'}$ can be formulated as follows:

$$\mathbf{K}^{'} = \begin{bmatrix} k_{1,1}^1 & \cdots & k_{1,n}^1 & k_{1,1}^2 & \cdots & k_{1,n}^2 \\ \vdots & \ddots & \vdots & \vdots & \ddots & \vdots \\ k_{n,1}^1 & \cdots & k_{n,n}^1 & k_{n,1}^2 & \cdots & k_{n,n}^2 \\ k_{1,1}^2 & \cdots & k_{1,n}^2 & k_{1,1}^1 & \cdots & k_{1,n}^1 \\ \vdots & \ddots & \vdots & \vdots & \ddots & \vdots \\ k_{n,1}^2 & \cdots & k_{n,n}^2 & k_{n,1}^1 & \cdots & k_{n,n}^1 \end{bmatrix}. \tag{13}$$

Analogous to equation 12, for $1 \leq i \leq n$, the probability of the $i$-th diversity-sensitive component in $\mathcal{D}^{'}$ being classified as category $c_i$ can be computed as follows:

$$\begin{aligned} \mathbf{P}^{'}[i,i] &= \frac{k_{i,i}^1}{\sum_j k_{i,j}^1 + \sum_j k_{i,j}^2} \\ &= \frac{k_{i,i}^1}{2\sum_j k_{i,j}^1} = \frac{k_{i,i}^1}{2\sum_j k_{i,j}^2} \\ &= \frac{1}{2}\mathbf{P}_1[i,i] = \frac{1}{2}\mathbf{P}_2[i,i]. \end{aligned} \tag{14}$$

For $n+1 \leq i \leq 2n$, we obtain the same result as depicted in equation 14. Consequently, the diversity of $\mathcal{D}^{'}$ can be computed as follows:

$$\begin{aligned} \text{DCScore}(\mathcal{D}^{'}) &= \sum_i^{2n} \mathbf{P}^{'}[i,i] \\ &= \frac{1}{2}\sum_i^n \mathbf{P}_1[i,i] + \frac{1}{2}\sum_i^n \mathbf{P}_2[i,i] \\ &= \sum_i^n \mathbf{P}_1[i,i] = \sum_i^n \mathbf{P}_2[i,i] \\ &= \text{DCScore}(\mathcal{D}_1) = \text{DCScore}(\mathcal{D}_2). \end{aligned} \tag{15}$$

$\square$

- **Symmetry (Restated)**: Diversity remains constant regardless of the order of the samples, exhibiting permutation invariance. Let $\pi$ denote the permutation function for the sample order, DCScore remains unchanged for any sample permutation of $\mathcal{D}$, i.e.,

$$\text{DCScore}(\mathcal{D}) = \text{DCScore}(\pi(\mathcal{D})). \tag{16}$$

*Proof.* According to equation 4, the order of samples does not affect the classification task. Thus, the diagonal elements of $\mathbf{P}$ remain unchanged, indicating the symmetry property of DCScore. $\square$

- **Monotonicity (Restated)**: The diversity of a dataset decreases as the similarity between its samples increases. Given two datasets $\mathcal{D}_1$ and $\mathcal{D}_2$, and a new sample $\mathcal{T}_{n+1}$, where the samples in $\mathcal{D}_1$ and $\mathcal{D}_2$ are entirely different, and $\text{DCScore}(\mathcal{D}_1) = \text{DCScore}(\mathcal{D}_2) = n$. If $\mathcal{T}_{n+1}$ is more similar to the samples in $\mathcal{D}_2$ than to those in $\mathcal{D}_1$ and is added to both datasets, then for the merged datasets $\mathcal{D}_1^{'}$ and $\mathcal{D}_2^{'}$, DCScore satisfies the following equation.

$$\text{DCScore}(\mathcal{D}_1^{'}) > \text{DCScore}(\mathcal{D}_2^{'}). \tag{17}$$

*Proof.* For $\mathcal{D}_1^{'} = \{\mathcal{T}_1^1, \mathcal{T}_2^1, ..., \mathcal{T}_n^1, \mathcal{T}_{n+1}\}$ and $\mathcal{D}_2^{'} = \{\mathcal{T}_1^2, \mathcal{T}_2^2, ..., \mathcal{T}_n^2, \mathcal{T}_{n+1}\}$, we have $\text{S}(\mathcal{T}_i^1, \mathcal{T}_{n+1}) < \text{S}(\mathcal{T}_j^2, \mathcal{T}_{n+1})$ for any $i,j = \{1,2,...,n\}$. Here, S is the similarity function. In this regard, the classification function $f$ exhibits lower confidence when classifying dataset $\mathcal{D}_2^{'}$, resulting in a lower probability that the $i$-th sample is classified into the $i$-th class, thereby leading to $\mathbf{P}_{\mathcal{D}_1^{'}}[i,i] > \mathbf{P}_{\mathcal{D}_2^{'}}[i,i]$. Then, the following formula is satisfied:

$$\mathbf{P}_{\mathcal{D}_1^{'}}[i,i] > \mathbf{P}_{\mathcal{D}_2^{'}}[i,i] \rightarrow \text{DCScore}(\mathcal{D}_1^{'}) > \text{DCScore}(\mathcal{D}_2^{'}), \tag{18}$$

where $\mathbf{P}_{\mathcal{D}_1^{'}}, \mathbf{P}_{\mathcal{D}_2^{'}}$ are the probability matrix of $\mathcal{D}_1^{'}, \mathcal{D}_2^{'}$, respectively. $\square$

# C    ALGORITHM

We further provide the PyTorch style pseudocode for implementing `DCScore`, as shown in algorithm 1. Our proposed method includes three stages: text representation, pairwise similarity, and diversity summarization. It is worth noting that the Softmax operation can be replaced with other Softmax like additive margin Softmax (Wang et al., 2018).

---

**Algorithm 1** PyTorch style pseudocode for `DCScore`.

```
# t: diversity-sensitive components within the evaluated dataset
# H: embeddings of t
# K: kernel matrix
# P: probability matrix

# text representation
H = embedder(t)

# pairwise similarity
K = kernel(H)

# diversity summarization
P = softmax(K)
DCScore = Trace(P)
```

---

# D    EXPERIMENTAL SETTINGS

## D.1    DATASETS

Two types of generated datasets, including our generated datasets and publicly available generated datasets, are employed in our experiments. We provide detailed information on these datasets below.

### D.1.1    OUR GENERATED DATASETS

We utilize two different our generated datasets in three subsections: Section 5.2.1, Section 5.2.2, and Section 5.4. It is worth noting that Section 5.2.1 and Section 5.4 share the same experimental datasets. We employ two commonly used LLMs as our dataset generator, including Llama2-13B (13B) and Llama2-70B (70B) (Touvron et al., 2023). To prompt LLMs to generate datasets, we design two prompts corresponding to *Zero-shot* and *Few-shot* generation settings, respectively. Additionally, our generated datasets involve two natural language generation (NLG) tasks: *text classification* and *story completion*. We set the maximum number of generated tokens to 100 and 30 for text classification and story completion tasks, respectively. The detailed generation information is offered as follows.

**Generation Settings.** We use different generation settings for generated datasets used in Section 5.2.1, Section 5.2.2, and Section 5.4.

- **Datasets on Section 5.2.1, Section 5.4, Appendix E.4, and Appendix E.5.** We generate 21 sub-datasets corresponding to different $\tau_g$ by varying $\tau_g$ from 0.2 to 1.2 with 0.05 intervals. For each sub-dataset, we employ LLMs (13B or 70B) to generate sets of 10 responses per context. Specifically, each sub-dataset consists of 100 samples.

- **Datasets on Section 5.2.2.** We employ the 70B model to generate 6 sub-datasets corresponding to different $\tau_g$ by varying $\tau_g$ from 0.2 to 1.2 with 0.2 intervals. Each sub-dataset includes 5 samples corresponding to a context. To repeat experiments five times, we use five different contexts to prompt the 70B model to generate 5 sub-datasets for each $\tau_g$.

- **Datasets on Appendix E.2.** In zero-shot or few-shot settings, we utilize the 70B model to generate three sub-datasets for the text classification task, corresponding to $\tau_g = \{0.2, 0.7, 1.2\}$, respectively. Unlike other settings that provide only one example, in this experiment, we adopt a few-shot setting where four examples and their corresponding labels are given, including two positive examples and two negative examples. Each sub-dataset

Table 5: Prompt settings for *zero-shot* and *few-shot* settings. Contents that need to be replaced are highlighted in gray .

| NLG Tasks | Zero-shot | Few-shot |
|---|---|---|
| **Text Classification** | Now you are a movie critic. You are given a movie genre/style and a length requirement. You must come up with a movie that corresponds to the genre/style and write a review that meets the length requirement. Write a film review for a {style} movie to express {pos_or_neg} feedback. Each review should have {num_of_words} words. Be sure to express your personal insights and feelings. Please be creative and write unique movie reviews. | Now you are a movie critic. You are given a movie genre/style and a length requirement. You must come up with a movie that corresponds to the genre/style and write a review that meets the length requirement. Write a film review according to the given example. Make sure your review expresses the same sentiment (positive or negative) as the example. Each review should have {num_of_words} words. Be sure to express your personal insights and feelings. Please be creative and write unique movie reviews. The following is the example: #An example from IMDB (Maas et al., 2011)# |
| **Story Completion** | Question: {story_q} Answer: | Complete the story according to the given example. Example: #An example from ROC Stories (Mostafazadeh et al., 2016)# Question: {story_q} Answer: |

contains 3,000 samples, and a context is employed to prompt the 70B model to generate five samples. To train text classification models on each sub-dataset, we randomly split 2,100 samples to the training set for each sub-dataset and gather the remaining 900 samples into the testing set across all three sub-datasets. Consequently, we construct a test set comprising 1,800 samples.

**Prompt Settings.** Following the setting of Li et al. (2023), we design different prompts for zero-shot and few-shot settings, respectively. For the text classification task under the zero-shot setting, we require LLMs to generate movie reviews with Sci-fi/Action/Drama/Comedy/Romance topics. Each movie review contains a single sentiment of either positive or negative, which is regarded as the text label. For the story completion task, we require LLMs to complete the story according to the given context. The detailed prompt setting is provided in Table 5.

In Table 5, "{style}" will be replaced with one topic within {Sci-fi, Action, Drama, Comedy, Romance} and "{pos_or_neg}" will be replaced with one label within {Positive, Negative}. "{num_of_words}" will be replaced with "50". "{story_q}" will be replaced by the first three sentences of each sample in the ROC Stories dataset.

### D.1.2 PUBLICLY AVAILABLE GENERATED DATASETS

We use SST2 (Socher et al., 2013), Yelp (Zhang et al., 2015), and AG News (Zhang et al., 2015), and their augmented version based on AttrPrompt (Yu et al., 2024). For three original datasets, we randomly sample data from training sets and apply this to the computational cost analysis in Appendix E.3. For three augmented datasets, each dataset has 6000 samples. We sample different

sub-datasets based on these three datasets, applied to Section 5.2.1 and Section 5.3, respectively. The details are as follows.

- **Datasets on Section 5.2.1.** We randomly sample 1000 samples from each of the three datasets.
- **Datasets on Section 5.3.** We remove samples with text token lengths less than 50 in the three datasets and then truncate each sample to a length of 50 tokens. Based on the above, we set up sub-datasets with randomly sampled samples of 100, 500, 1000, 2000, and 4000 according to experimental settings.

### D.2 IMPLEMENTATION DETAILS

For three transformation-based methods, including `DCScore`, *VendiScore*, and *K-means Inertia*, we employ *unsup-simcse-bert-base-uncased* (princeton nlp, 2021) as the embedding function. For all dataset generation language models, we set the top-p and top-k parameters to 1 and -1, respectively. Additionally, we limit the maximum number of newly generated tokens to 100 for the text classification task and 30 for the story completion task. All experiments are conducted on $8\times$ NVIDIA Tesla V100 GPU with 32GB memory.

#### D.2.1 HYPERPARAMETER SETTINGS OF DIVERSITY EVALUATION

For `DCScore`, *VendiScore*, and *K-means Inertia*, we fix the batch size of generating sample representation at 128 across all experiments. Given the varying hyperparameters for each diversity evaluation method, we provide the detailed settings for each method below:

- **DCScore.** We employ the cosine similarity as $\text{Kernel}(\cdot)$, and Softmax as $f_{\mathbf{K}}(\cdot)$. Except for hyperparameter sensitivity experiments, we set $\tau$ in equation 4 to 1 for all other experiments.
- **Distinct-n.** We use 5-grams to calculate distinct-n.
- **K-means Inertia.** We set the number of clusters to 10 for all experiments.
- **VendiScore.** We employ the cosine similarity as $\text{Kernel}(\cdot)$.

#### D.2.2 HYPERPARAMETER SETTINGS OF DOWNSTREAM TASK TRAINING

To train text classification models, we employ RoBERTa (Liu, 2019) as the encoder and utilize the representations from the last layer of the encoder as the classifier's input. We employ LoRA (Hu et al., 2021) to finetune the encoder and the classifier on our generated datasets. Specifically, we fix the LoRA scaling factor to 32 and the rank of the update matrices to 8. We use AdamW (Loshchilov, 2017) with an initial learning rate of 5e-5 and linear learning rate decay as our optimizer. Additionally, we set the batch size per GPU as 32 and epochs as 120. For the number of training GPUs, we employ 8 GPUs for zero-shot settings and 4 GPUs for few-shot settings. Therefore, the different training steps for zero-shot and few-shot settings are shown in Figure 8.

#### D.2.3 EVALUATION PROTOCOL

In our experiments, we employ diversity evaluation methods to score the diversity of sub-datasets using two evaluation protocols: *overall evaluation* and *batch evaluation*. While *K-means Inertia* uses the overall evaluation protocol, all other methods utilize the batch evaluation protocol. The detailed settings for the two evaluation protocols are as follows:

- **Batch evaluation.** Due to a context or prompt associated with several samples in a sub-dataset, the batch evaluation protocol requires that evaluation methods treat samples generated from the same context as a single batch. The evaluation results are then averaged across all batches of the entire sub-dataset.
- **Overall evaluation.** We consider each sample in a sub-dataset as independent, meaning each sample is generated by a distinct context or prompt. Based on this assumption, the overall evaluation protocol requires evaluation methods to directly measure the diversity of the entire sub-dataset.

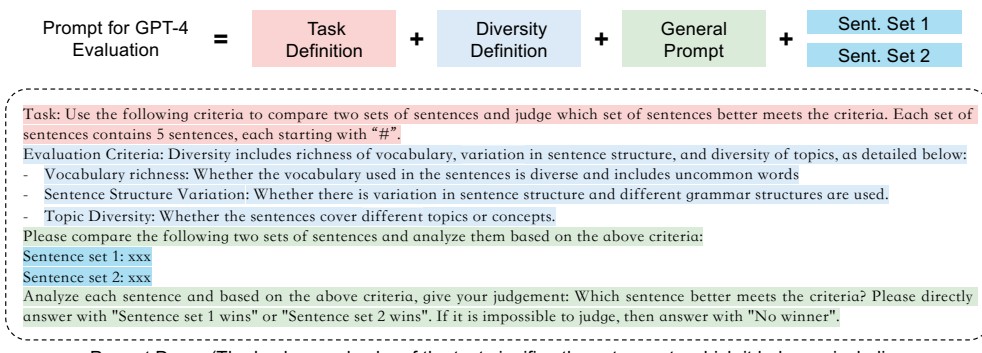

Prompt Demo (The background color of the text signifies the category to which it belongs, including the task definition, diversity definition, general prompt, and sentence sets.)

Figure 7: Prompt settings for GPT-4 evaluations.

Table 6: Pairwise correlation (Spearman's $\rho$) between GPT-4, temperature ($\tau_g$), and DCScore. DCScore indicates a strong correlation with GPT-4 evaluation results.

|  | Story-Few | Story-Zero | Text-Few | Text-Zero |
|---|---|---|---|---|
| GPT-4-DCScore | $0.6057_{\pm 0.30}$ | $0.9010_{\pm 0.04}$ | $0.6131_{\pm 0.18}$ | $0.9052_{\pm 0.09}$ |
| $\tau_g$-DCScore | $0.6757_{\pm 0.30}$ | $0.8782_{\pm 0.08}$ | $0.5714_{\pm 0.27}$ | $0.9336_{\pm 0.06}$ |
| $\tau_g$-GPT-4 | $0.9086_{\pm 0.07}$ | $0.7829_{\pm 0.16}$ | $0.8400_{\pm 0.16}$ | $0.8971_{\pm 0.07}$ |

### D.2.4 PROMPT SETTINGS OF LLM EVALUATION

In Section 6, we use GPT-4 to perform pairwise diversity comparisons. To guide GPT-4 in making these comparisons, we employ a well-designed prompt, as illustrated in Figure 7. The prompt for GPT-4 evaluations includes the task definition, diversity definition, general prompt, and sentence sets to be compared.

## E  ADDITIONAL EXPERIMENTS

### E.1  CORRELATION WITH LLM EVALUATOR

To further verify the effectiveness of DCScore, we investigate the evaluation correlation between DCScore and LLMs. Following the setting in Section 5.2.2, we employ GPT-4 to conduct pairwise comparisons between two generated datasets with different $\tau_g$. It is worth noting that these generated datasets are identical to those used in Section 5.2.2. Based on the pairwise comparison results, we obtain the diversity ranking outcomes. Regarding GPT-4 evaluation results as the diversity pseudo-truth, we report the pairwise evaluation correlation between DCScore, GPT-4, and $\tau_g$ in Table 6. We observe that DCScore exhibits strong correlations with GPT-4 and $\tau_g$ in zero-shot settings. By comparing the results of "$\tau_g$-DCScore" and "$\tau_g$-GPT-4", we find that DCScore outperforms the GPT-4 evaluator in terms of correlation with $\tau_g$ in zero-shot settings. Regarding the correlation performance in few-shot settings, we notice lower correlations of all baseline methods compared to zero-shot settings. We guess that this phenomenon is related to the distributions of the generated datasets. Although DCScore exhibits lower correlations (about 0.6) with GPT-4, this result can still be considered a strong correlation according to Akoglu (2018).

### E.2  CORRELATION WITH DOWNSTREAM TASK TRAINING

To investigate the correlation between DCScore and downstream task training, we train text classification models using our generated datasets under zero-shot and few-shot settings. We vary the generation temperature $\tau_g$ of our generated datasets within the range of $\{0.2, 0.7, 1.2\}$. More details of training datasets and hyperparameters are presented in Appendix D.1.1 and D.2.2, respectively. Figure 8 shows the loss curves of these trained classification models. In the zero-shot setting, we

Table 7: Downstream task training performance and diversity evaluation on our generated datasets with $\tau_g = \{0.2, 0.7, 1.2\}$.

| | Accuracy | | | DCScore | | |
| | $\tau_g$=0.2 | $\tau_g$=0.7 | $\tau_g$=1.2 | $\tau_g$=0.2 | $\tau_g$=0.7 | $\tau_g$=1.2 |
|---|---|---|---|---|---|---|
| **Zero-shot** | 89.10 | 89.70 | 90.37 | 481.76 | 1745.42 | 2082.42 |
| **Few-shot** | 70.07 | 73.19 | 72.44 | 1376.43 | 1958.16 | 2047.90 |

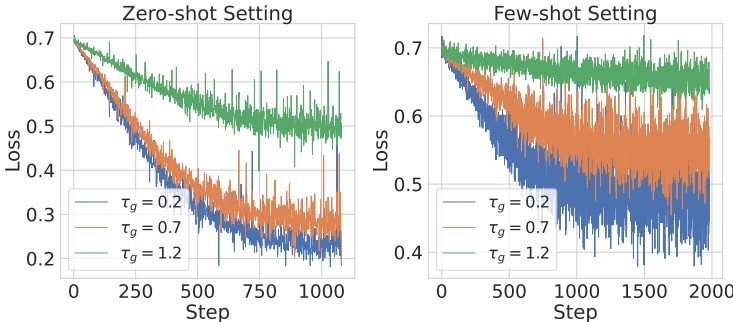

Figure 8: Loss curves of the downstream task training.

observe increasing optimal loss values as $\tau_g$ varied from 0.2 to 1.2, indicating that the model is more easily fitted to datasets with limited diversity. However, as shown in Table 7, models trained on more diverse datasets achieve better accuracy, which can be attributed to their enhanced generalization capabilities. From Table 7, the diversity evaluated by DCScore has a similar trend to the accuracy performance in the zero-shot setting, further demonstrating the effectiveness of DCScore in diversity evaluation.

In the few-shot setting, we observe a trend in optimal loss variation similar to that in the zero-shot setting, as shown in Figure 8. However, as shown in Table 7, the performance of the model trained on the dataset generated with $\tau_g = 1.2$ is inferior to that of the model trained on the dataset generated with $\tau_g = 0.7$, which contrasts with the findings in the zero-shot setting. This phenomenon can be attributed to the higher diversity of datasets generated at a higher $\tau_g$, resulting in increased fitting difficulty. Under the current settings, the number of training epochs for the dataset generated at a temperature of 1.2 is insufficient, preventing the trained model from achieving optimal performance. To validate this hypothesis, we increase the number of epochs to 240 and 360 and train models on the dataset generated at a temperature of 1.2. The final training loss and accuracy of these models are shown in Figure 9. We observe that as the number of epochs increases, the model's loss gradually decreases, and its performance improves progressively. Ultimately, the model's accuracy outperforms that of models trained on datasets generated at temperatures of 0.2 and 0.7. Moreover, from Table 7, models trained on datasets from the zero-shot setting outperform those trained on datasets from the few-shot setting. However, this discrepancy arises from the different test sets used in the two settings, making direct performance comparisons inappropriate.

### E.3 COMPUTATIONAL COST FOR LARGER DATASETS

In the case where the inner product is used as the kernel function and $n \gg d$, *VendiScore* can significantly reduce computational complexity. To ensure a fair comparison, we compare the computation times of *VendiScore* and DCScore on larger datasets, as well as under different kernel functions. Specifically, we employ SST2, Yelp, and AG News as the evaluated datasets. We randomly sample 4k, 8k, 16k, 32k, 64k samples and record the computation times for both methods across these different sample sizes. We repeat the experiments 5 times to report the mean and standard deviation.

As shown in Tables 8-10, DCScore has a shorter computation time than *VendiScore* in most cases, with *VendiScore* only exhibiting a computational advantage when the inner product is used and $n \gg d$. Furthermore, as the sample size increases, the efficiency advantage of DCScore becomes more pronounced. When using a polynomial kernel, on the SST2 dataset, DCScore requires only one-

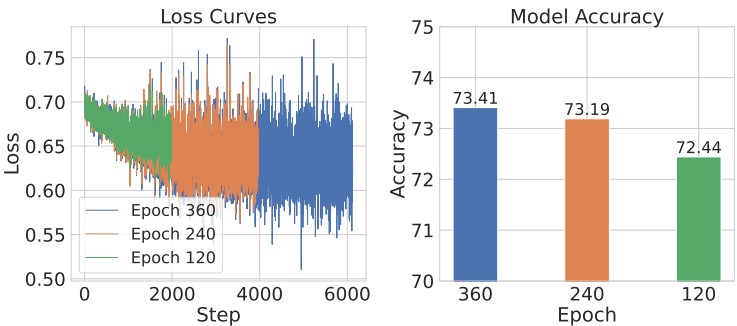

Figure 9: Loss curves and accuracy of models trained on generated dataset with $\tau_g = 1.2$.

Table 8: Comparison of computation time between `DCScore` and VendiScore on SST2.

| Kernels | | SST2 | | | | |
|---|---|---|---|---|---|---|
| | Sample num | 4k | 8k | 16k | 32k | 64k |
| Inner product | VendiScore | $4.65_{\pm 0.28}$ | $\mathbf{9.84}_{\pm 0.26}$ | $\mathbf{19.02}_{\pm 0.70}$ | $\mathbf{37.31}_{\pm 1.88}$ | $\mathbf{76.19}_{\pm 1.91}$ |
| | DCScore | $\mathbf{4.58}_{\pm 0.29}$ | $10.03_{\pm 0.17}$ | $20.42_{\pm 0.39}$ | $42.91_{\pm 1.59}$ | $112.47_{\pm 2.43}$ |
| RBF kernel | VendiScore | $5.86_{\pm 0.06}$ | $12.41_{\pm 0.49}$ | $32.94_{\pm 0.40}$ | $100.36_{\pm 1.44}$ | $449.14_{\pm 10.35}$ |
| | DCScore | $\mathbf{5.22}_{\pm 0.33}$ | $\mathbf{9.94}_{\pm 0.42}$ | $\mathbf{21.20}_{\pm 0.75}$ | $\mathbf{46.57}_{\pm 1.47}$ | $\mathbf{117.06}_{\pm 1.91}$ |
| Poly kernel | VendiScore | $5.73_{\pm 0.06}$ | $12.72_{\pm 0.41}$ | $31.47_{\pm 0.97}$ | $98.31_{\pm 0.25}$ | $453.11_{\pm 2.53}$ |
| | DCScore | $\mathbf{5.09}_{\pm 0.28}$ | $\mathbf{10.27}_{\pm 0.12}$ | $\mathbf{20.12}_{\pm 1.02}$ | $\mathbf{46.25}_{\pm 1.82}$ | $\mathbf{123.51}_{\pm 3.40}$ |

Table 9: Comparison of computation time between `DCScore` and VendiScore on Yelp.

| Kernels | | Yelp | | | | |
|---|---|---|---|---|---|---|
| | Sample num | 4k | 8k | 16k | 32k | 64k |
| Inner product | VendiScore | $57.96_{\pm 0.35}$ | $\mathbf{114.64}_{\pm 1.63}$ | $\mathbf{227.76}_{\pm 7.04}$ | $\mathbf{451.49}_{\pm 19.73}$ | $\mathbf{912.60}_{\pm 25.69}$ |
| | DCScore | $\mathbf{57.95}_{\pm 0.31}$ | $115.35_{\pm 1.16}$ | $232.49_{\pm 1.34}$ | $448.98_{\pm 23.94}$ | $961.29_{\pm 2.86}$ |
| RBF kernel | VendiScore | $59.31_{\pm 0.06}$ | $118.15_{\pm 0.91}$ | $242.06_{\pm 7.60}$ | $527.99_{\pm 2.89}$ | $1272.93_{\pm 21.15}$ |
| | DCScore | $\mathbf{58.49}_{\pm 0.14}$ | $\mathbf{116.29}_{\pm 0.92}$ | $\mathbf{232.94}_{\pm 3.09}$ | $\mathbf{471.18}_{\pm 7.80}$ | $\mathbf{953.62}_{\pm 17.21}$ |
| Poly kernel | VendiScore | $59.48_{\pm 0.05}$ | $118.94_{\pm 0.95}$ | $234.08_{\pm 11.72}$ | $522.82_{\pm 3.04}$ | $1313.55_{\pm 12.64}$ |
| | DCScore | $\mathbf{58.73}_{\pm 0.08}$ | $\mathbf{117.02}_{\pm 0.90}$ | $\mathbf{227.72}_{\pm 9.51}$ | $\mathbf{462.45}_{\pm 13.91}$ | $\mathbf{988.53}_{\pm 1.10}$ |

third of the computation time of *VendiScore* when the sample size reaches 64k. In contrast, although *VendiScore* has a computational advantage in the case of the inner product, the difference compared to `DCScore` is not significant. The experimental results are consistent with our complexity analysis presented in Appendix F. Overall, `DCScore` outperforms *VendiScore* in terms of computation time across most kernel functions. *VendiScore* exhibits a computational time advantage only when the inner product is used as the kernel function, which will limit its applicability. As shown in Chapter 4 of Seeger (2004), it is essential to employ different kernel functions to accommodate a wider range of scenarios.

### E.4 IMPACT OF EMBEDDING FUNCTIONS Φ

The paradigm of the transformation-based method enables `DCScore` to utilize various embedding functions tailored to different scenarios. Consequently, we investigate the impact of embedding functions on our generated datasets used in Section 5.2.1. As shown in Table 11, we compare the correlation of the diversity evaluation results of `DCScore` across 4 different embedding functions with diversity pseudo-truths, where the model names in parentheses within the embedding function refer to those available on Hugging Face. Our findings indicate that `DCScore` exhibits strong correlations with diversity pseudo-truths across various embedding functions. Notably, `DCScore`

Table 10: Comparison of computation time between `DCScore` and VendiScore on AG News.

| Kernels | AG News | | | | | |
|---|---|---|---|---|---|---|
| | Sample num | 4k | 8k | 16k | 32k | 64k |
| Inner product | VendiScore | $14.56_{\pm1.16}$ | $30.20_{\pm1.15}$ | $63.70_{\pm1.39}$ | $127.25_{\pm1.13}$ | $254.20_{\pm11.71}$ |
| | DCScore | $14.61_{\pm1.15}$ | $30.61_{\pm1.77}$ | $63.57_{\pm2.68}$ | $129.70_{\pm4.17}$ | $284.76_{\pm12.30}$ |
| RBF kernel | VendiScore | $16.69_{\pm1.54}$ | $33.69_{\pm1.47}$ | $80.09_{\pm2.34}$ | $185.79_{\pm6.44}$ | $617.06_{\pm12.51}$ |
| | DCScore | $16.01_{\pm1.53}$ | $31.06_{\pm0.96}$ | $69.15_{\pm1.32}$ | $129.36_{\pm5.56}$ | $297.29_{\pm3.67}$ |
| Poly kernel | VendiScore | $17.60_{\pm0.62}$ | $36.16_{\pm1.27}$ | $79.34_{\pm1.57}$ | $190.96_{\pm2.75}$ | $632.69_{\pm10.14}$ |
| | DCScore | $16.88_{\pm0.59}$ | $33.78_{\pm1.28}$ | $68.18_{\pm1.66}$ | $138.18_{\pm3.82}$ | $303.06_{\pm11.40}$ |

Table 11: Correlation (Spearman's $\rho$) results of `DCScore` with various embedding models. Spearman's $\rho$ varies between -1 and +1 with 0 implying no correlation. Best results are indicated in **bold**.

| Embedding models | Zero-shot setting | | | | Few-shot setting | | | |
|---|---|---|---|---|---|---|---|---|
| | Text classification | | Story completion | | Text classification | | Story completion | |
| | 13B | 70B | 13B | 70B | 13B | 70B | 13B | 70B |
| SimCSE (unsup-simcse-bert-base-uncased) | **0.9961** | 0.9779 | 0.9844 | 0.9792 | **0.9909** | 0.9883 | 0.9857 | **0.9974** |
| SimCSE (sup-simcse-roberta-large) | 0.9909 | 0.9753 | 0.9883 | 0.9883 | 0.9792 | **0.9935** | 0.9779 | 0.9623 |
| Sentence BERT (all-mpnet-base-v2) | 0.9896 | 0.9740 | 0.9870 | 0.9909 | 0.9766 | 0.9870 | 0.9857 | 0.9870 |
| BGE (bge-large-en-v1.5) | 0.9909 | **0.9896** | **0.9922** | **0.9948** | 0.9857 | 0.9922 | **0.9870** | 0.9922 |

Table 12: Correlation (Spearman's $\rho$) results of `DCScore` with various kernel functions. Spearman's $\rho$ varies between -1 and +1 with 0 implying no correlation. Best results are indicated in **bold**.

| Embedding models | Zero-shot setting | | | | Few-shot setting | | | |
|---|---|---|---|---|---|---|---|---|
| | Text classification | | Story completion | | Text classification | | Story completion | |
| | 13B | 70B | 13B | 70B | 13B | 70B | 13B | 70B |
| **Inner product** | **0.9961** | 0.9779 | 0.9844 | **0.9792** | **0.9909** | **0.9883** | 0.9857 | **0.9974** |
| **laplacian kernel** | 0.9935 | **0.9831** | 0.9883 | 0.9727 | 0.9597 | 0.9649 | 0.9701 | 0.9922 |
| **RBF kernel** | 0.9935 | 0.9818 | **0.9896** | 0.9753 | 0.9740 | 0.9727 | 0.9792 | 0.9922 |
| **polynomial kernel** | 0.9870 | 0.9584 | 0.9714 | 0.9506 | 0.9182 | 0.9182 | 0.9857 | 0.9896 |

utilizing the BGE embedding function achieves the best results in half of the cases. Additionally, the minimum correlation in Table 11 exceeds 0.96, which is classified as a strong correlation according to Akoglu (2018). This result also supports the following two conclusions: (1) the embedding function used effectively captures the differences among samples in the dataset from multiple perspectives, and (2) `DCScore` is sufficiently adaptable to different embedding functions while maintaining stable performance.

### E.5 IMPACT OF KERNEL FUNCTIONS

Similar to Appendix E.4, we investigate the impact of different kernel functions on the performance of `DCScore`. Specifically, this experimental setup is identical to that in Appendix E.4. As shown in Table 12, we find that `DCScore` demonstrates stable performance across various kernel functions. However, the influence of the kernel function is slightly more pronounced than that of the embedding function, as indicated by the greater fluctuations in correlation among the different kernel functions. Furthermore, we observe that `DCScore` achieves optimal performance in the case of the inner product. Overall, `DCScore` consistently maintains strong diversity evaluation performance across different kernel functions.

Table 13: Comparison of Complexity analysis between DCScore and VendiScore. $\mathcal{O}_{kernel}$ represents the complexity of the kernel function.

|  |  | General Kernels | Inner Product |
|---|---|---|---|
| **Pairwise Similarity** | **Vendi Score** | $\mathcal{O}(n^2 \cdot \mathcal{O}_{kernel})$ | $\mathcal{O}(d^2 n)$ |
|  | **DCScore** |  | $\mathcal{O}(n^2 d)$ |
| **Summarization** | **Vendi Score** | $\mathcal{O}(n^3)$ | $\mathcal{O}(d^3)$ |
|  | **DCScore** | $\mathcal{O}(n^2)$ | $\mathcal{O}(n^2)$ |
| **Total** | **Vendi Score** | $\mathcal{O}(n^2 \cdot \mathcal{O}_{kernel} + n^3)$ | $\mathcal{O}(d^2 n + d^3) = \mathcal{O}(d^2 n)$ |
|  | **DCScore** | $\mathcal{O}(n^2 \cdot \mathcal{O}_{kernel} + n^2)$ | $\mathcal{O}(n^2 d + n^2)$ |

## F  COMPLEXITY ANALYSIS

To enhance our understanding of DCScore, we provide a brief analysis of its time complexity in Table 13. Denoting $\mathcal{O}_{kernel}$ as the complexity associated with general kernels (i.e., kernels other than linear kernels), we analyze the complexity in the pairwise similarity and summarization stages. In the pairwise similarity stage, the computation of pairwise similarities results in a complexity of $\mathcal{O}(n^2)$ for DCScore. When combined with the complexity $\mathcal{O}_{kernel}$ of the general kernel computation, DCScore exhibits a total complexity of $\mathcal{O}(n^2 \cdot \mathcal{O}_{kernel})$ in this stage. In the summarization stage, DCScore has a complexity of $\mathbf{O}(n^2)$ due to the softmax operation. Consequently, the overall complexity of DCScore for general kernels is $\mathcal{O}(n^2 \cdot \mathcal{O}_{kernel} + n^2)$. In contrast, *VendiScore* has a total complexity of $\mathcal{O}(n^2 \cdot \mathcal{O}_{kernel} + n^3)$, where the pairwise similarity stage is identical to that of DCScore, while the summarization stage incurs a complexity of $\mathcal{O}(n^3)$ due to the eigenvalue computation. Thus, for general kernels, DCScore demonstrates lower complexity than *VendiScore*.

However, when the inner product is employed as the kernel function and $n \gg d$, *VendiScore* can significantly reduce the complexity by replacing the pairwise similarity $XX^T$ with $X^T X$, where $X \in \mathbb{R}^{n \times d}$. This results in complexities of $\mathcal{O}(d^2 n)$ for the pairwise similarity stage and $\mathcal{O}(d^3)$ for the summarization stage. In this regard, DCScore has a complexity of $\mathcal{O}(n^2 d + n^2)$, which is slightly worse than that of *VendiScore*. Fortunately, we can leverage efficient techniques, such as those proposed in Shim et al. (2017) and Wen et al. (2023), to reduce the computational complexity of DCScore. Overall, compared to *VendiScore*, DCScore maintains lower complexity in most cases, as empirically validated in Section 5.3 and Appendix E.3. Although *VendiScore* has a lower complexity when the inner product is used as the kernel, experiments in the Appendix E.3 show that the computation times of *VendiScore* and DCScore are quite similar. Additionally, in practical applications, different kernels can maintain low computational complexity, which is more useful than being restricted to a single kernel.

## G  DETAILED MODELING OF EXISTING METHODS

We present the detailed modeling of existing methods into DCScore as follows:

**Distinct-n.** *Distinct-n* (Li et al., 2015) is a prevalent diversity metric depending on n-grams, where *n* signifies the number of successive items. *Distinct-n* calculates the proportion of unique n-grams to all n-grams. The n-grams operation falls under the text representation stage, while the step of obtaining a unique set of n-grams corresponds to the pairwise similarity stage. Typically, a high form similarity among samples in the evaluated dataset results in a smaller unique n-grams set. Finally, ratio calculations belong to the diversity summarization stage.

$$\textit{Text Representation: } \text{n-grams}(\text{Concat}(\mathcal{D})),$$
$$\textit{Pairwise Similarity: } \text{Unique}(\text{n-grams}(\text{Concat}(\mathcal{D}))),$$
$$\textit{Diversity Summarization: } \text{Distinct-n}(\mathcal{D}) = \frac{|\text{Unique}(\text{n-grams}(\text{Concat}(\mathcal{D})))|}{|\text{n-grams}(\text{Concat}(\mathcal{D}))|}, \tag{19}$$

where n-grams, Unique, and Concat represent the n-grams, de-overlap process, and concatenate operation, respectively.

**K-means inertia.** *K-means inertia* (Du & Black, 2019), a transformation-based method, performs clustering in sample representation and then calculates inertia as diversity outcomes. Here, inertia refers to the square summarization of the distance between samples and cluster centroids.

$$
\begin{aligned}
&\textit{Text Representation: } \mathbf{H} = \Phi(\{\tilde{\mathcal{T}}_i\}_{i=1}^n), \\
&\textit{Pairwise Similarity: } \mathcal{C} = \text{K-means}(\mathbf{H}), \\
&\textit{Diversity Summarization: } \text{Inertia}(\mathcal{D}) = \sum_{\mathbf{c}_k \in \mathcal{C}, \mathbf{h}_j \in \mathbf{H}_{\mathbf{c}_k}} (\mathbf{h}_j - \mathbf{c}_k)^2,
\end{aligned}
\tag{20}
$$

where $\mathbf{H}$ is the representation of all samples and $\mathbf{h}_i$ is the representation of the $i$-th sample, $\mathcal{C}$ denotes the cluster centroid set, and $\mathbf{c}_k \in \mathcal{C}$ represents the $k$-th cluster centroid. The sample representation associated with the $k$-th cluster centroid is expressed as $\mathbf{h}_j \in \mathbf{H}_{\mathbf{c}_k}$, while $\mathbf{H}_{\mathbf{c}_k}$ denotes the sample representations within the $k$-th cluster centroid.

**VendiScore.** *VendiScore* (Dan Friedman & Dieng, 2023) is a recently proposed diversity metric that falls under the category of the transformation-based method. Based on sample representations, VendiScore utilizes a kernel function to calculate similarity matrix $\mathbf{K}$. Subsequently, VendiScore summarizes diversity as the exponential of the Shannon entropy of the eigenvalues of $\mathbf{K}/n$.

$$
\begin{aligned}
&\textit{Text Representation: } \mathbf{H} = \Phi(\{\tilde{\mathcal{T}}_i\}_{i=1}^n), \\
&\textit{Pairwise Similarity: } \mathbf{K} = \text{Kernel}(\mathbf{H}), \\
&\textit{Diversity Summarization: } \text{VS}(\mathcal{D}) = \exp\big(-\sum_{i=1}^n \lambda_i \log \lambda_i\big),
\end{aligned}
\tag{21}
$$

where $\text{Kernel}(\cdot)$ is the kernel function, such as the cosine similarity, $\lambda_i$ is the $i$-th eigenvalue of $\mathbf{K}/n$.

## H  OTHER IMPLEMENTATIONS FOR A CLASSIFICATION PERSPECTIVE

Equation 9 represents one way to implement the classification perspective of DCScore. There are other potential implementations of DCScore with lower complexity. When $n \gg d$, the classification probability modeling for any sample within the evaluated dataset is primarily determined by samples that are relatively similar to it, while the majority of other samples fall into the tail probability category. Based on this observation, we can reduce computational cost by calculating pairwise similarity only among similar samples. Therefore, we propose a feasible approach named DCScore$_{cluster}$ below.

DCScore$_{cluster}$ defines similar samples as those within the same cluster. Specifically, DCScore$_{cluster}$ involves obtaining cluster centers and the membership of each sample point through clustering, followed by calculating pairwise similarities among samples within the same cluster. By performing clustering, we identify similar samples for each sample point to facilitate pairwise similarity calculation, thereby avoiding the computation of global pairwise similarities and further reducing the computational complexity of the method. Based on this process, the method can be formulated as follows:

$$
\begin{aligned}
&\textit{Text Representation: } \mathbf{H} = \Phi(\{\tilde{\mathcal{T}}_i\}_{i=1}^n), \\
&\textit{Pairwise Similarity: } \mathcal{C} = \text{K-means}(\mathbf{H}), \\
&\mathbf{K}[i,j] = \begin{cases} \text{Kernel}(\mathbf{h}_i, \mathbf{h}_j), & \text{if } \mathbf{h}_i, \mathbf{h}_j \in \mathbf{C}_k \\ 0, & \text{if } \mathbf{h}_i \in \mathbf{C}_k \text{ and } \mathbf{h}_j \in \mathbf{C}_l \text{ with } k \neq l \end{cases} \\
&\textit{Diversity Summarization: } \mathbf{P}[i,j] = \begin{cases} f_{\mathbf{K}}(\mathbf{K}[i,j]), & \text{if } \mathbf{h}_i, \mathbf{h}_j \in \mathbf{C}_k \\ 0, & \text{if } \mathbf{h}_i \in \mathbf{C}_k \text{ and } \mathbf{h}_j \in \mathbf{C}_l \text{ with } k \neq l \end{cases} \\
&\text{DCScore}(\mathcal{D}) = \text{tr}(\mathbf{P}) = \sum_{i=1}^n \mathbf{P}[i,i].
\end{aligned}
\tag{22}
$$

where $\mathbf{C}_k, \mathbf{C}_l \in \mathcal{C}$ are clusters.

