# OpenReview forum: "Evaluating Diversity of LLM-generated Datasets: A Classification Perspective"
_ICLR.cc/2025/Conference — Submitted to ICLR 2025_

### Official Review · Reviewer_fedN · 2024-11-02

**Soundness:** 2
**Presentation:** 3
**Contribution:** 3
**Rating:** 5
**Confidence:** 4

**Summary:**

This work proposes DCScore to evaluate the diversity of LLM-generated datasets.DCScore treats the diversity evaluation as a sample classification task, considering mutual relationships among samples. The proposed method is somewhat novel to me.

**Strengths:**

The proposed method can be used to evaluate the diversity of LLM-generated datasets and capture the mutual relationships among samples.
Extensive experiments show that DCScore exhibits a stronger correlation with diversity parameters and human judgment than baseline metrics.
The authors evaluate both the performance and efficiency of DCScore, highlighting its effectiveness.

**Weaknesses:**

1. Some related works [1,2] evaluate the quality of augmented data from the data augmentation perspective. They also involve diversity-related metrics. It is suggested that authors discuss and compare with them.
[1] Gontijo-Lopes, Raphael, et al. "Affinity and diversity: Quantifying mechanisms of data augmentation." arXiv preprint arXiv:2002.08973 (2020).
[2] Yang, Suorong, et al. "Investigating the effectiveness of data augmentation from similarity and diversity: An empirical study." Pattern Recognition 148 (2024): 110204.
2. The diversity-sensitive components are important for DCScore calculations. How are these components computed? Since this is one of the most critical parts of the proposed method,  more specific details are needed.
3. I got a little confused about the definition and properties of DCScore. If I understand correctly, in the probability matrix P, P[i,i] denotes the probability of classifying T_i into class I. 1. What is the meaning of categories in the classification function? 2. If all samples are identical, what does the DCScore equal? If this is 1, the probability is one-hot and contracts to softmax calculations. If not, then the DCScore does not equal 1. The same is true for cases where all samples are distinct. More clarifications are needed.
4. On page 8, evaluation of existing datasets. How can we evaluate the effectiveness of different metrics? Are some datasets more diverse than others, and can the proposed metrics reflect this trend?
5. Is the diversity the higher, the better?  Can authors provide some takeaway conclusions for readers?
I will increase my score based on the authors' responses.

**Questions:**

Please see the weakness.

---

> ### Author Response · Authors · 2024-11-21
> **Response to Reviewer fedN (part 1/2)**
>
> We appreciate the reviewer's detailed comments and suggestive feedback. We want to clarify some misunderstandings that caused some of your concerns. **We would appreciate it if you could consider our responses when making the final evaluation of our work. Our revised manuscript will be uploaded later, and the updated content is highlighted in blue.**
>
> ---
> > Q1: Some related works [1,2] evaluate the quality of augmented data from the data augmentation perspective. They also involve diversity-related metrics. It is suggested that authors discuss and compare with them.
>
> **R1:** Thank you for your suggestion. The discussion and comparison between these two works are summarized as follows, and **we will consider incorporating these discussions into the related work section in a future version.**
>
> **Comparison with related work 1:** Related work 1 proposes three diversity metrics to quantify augmentation performance, which can be named final training loss, conditional entropy, and training time. The differences between DCScore and this work are as follows:
>
> - **Different purposes:** DCScore aims to assess the diversity of datasets, while Related Work 1 focuses on exploring the effectiveness of data augmentation methods from the perspective of the diversity of augmented samples.
>
> - **Different physical meanings:** The evaluation results obtained from DCScore can be understood in terms of the number of effective samples in the dataset, which is consistent with the Vendi Score. In contrast, the three diversity evaluation metrics in Related Work 1 do not provide any physical meaning.
>
> - **Independence from training models:** DCScore focuses solely on the evaluation of the dataset itself and is independent of the models trained on the dataset. In Related Work 1, both the calculation of the final training loss and training time metrics are associated with the training model.
>
> - **Different computational complexities:** DCScore evaluates dataset diversity without involving model training, resulting in lower complexity. In contrast, the final training loss and training time in Related Work 1 both involve the model training process, which has a high level of complexity.
>
> **Comparison with related work 2:** Related Work 2 also aims to evaluate the effectiveness of data augmentation via diversity metrics. The differences between DCScore and this work are as follows:
>
> - **Different purposes:** Related Work 2 primarily explores the effectiveness of data augmentation methods through diversity evaluation.
>
> - **Different universality:** The method in Related Work 2 can be understood as a special case of the Vendi Score, limited to using only the inner product for similarity calculation. After calculating the eigenvalues and eigenvectors in the same way as the Vendi Score, Related Work 2 defines diversity based on the differences in eigenvalues and eigenvectors between the original data and the augmented data. This method's calculations are constrained by the similarity calculation approach and the original dataset. In contrast, DCScore overcomes these limitations, offering better universality.
>
> In addition to the above two discussions, DCScore also differs in its definition of diversity from the two aforementioned works. **DCScore defines diversity more in terms of the distinguishability among samples in the dataset, while the two related works consider diversity to be more about the inherent complexity of the data itself.**
>
> > Q2: The diversity-sensitive components are important for DCScore calculations. How are these components computed?
>
> **R2:** The explanation of the diversity-sensitive components was already indicated in Section 3.2. **The diversity-sensitive components vary depending on the downstream task and do not require calculation**. Specifically, **the diversity-sensitive components refer to the parts of the dataset samples whose diversity affects the performance of the trained model.** For example, in the context of text classification tasks involving movie reviews, each sample in the dataset consists of a movie review $Q$ and its corresponding sentiment label $A$. In this case, $A$ may simply be a numerical value (0 or 1), meaning that only the diversity of $Q$ affects the performance of the trained model. Therefore, in this scenario, $Q$ is the diversity-sensitive component. In our revised manuscript, we further improve the writing of the diversity-sensitive components in Section 3.2.

---

> ### Author Response · Authors · 2024-11-21
> **Response to Reviewer fedN (part 2/2)**
>
> > Q3: I got a little confused about the definition and properties of DCScore.
> (1) What is the meaning of categories in the classification function?
> (2) If all samples are identical, what does the DCScore equal?
>
> **R3:** For (1), **the categories of the classification function correspond to the abstract classes of each sample in the evaluated dataset.** For example, when the dataset to be evaluated has $n$ samples, the number of categories in the classification function will be $n$, with each category corresponding to each sample. For the probability matrix $P$, $P[i,j]$ represents the probability of sample $i$ being classified into class $j$, which can also be understood as the similarity between sample $i$ and sample $j$.
>
> For (2), when each sample is equal, **the DCScore will equal 1**, which also implies that there is only one effective sample in the dataset. We have provided an explanation in the previous version of our paper in Section 4.2 (Effective Number) and included proof in Appendix B. According to Eq. 5, DCScore can be summarized as the trace of the probability matrix, which means that **each diagonal element of the probability matrix is $1/n$.**
>
> > Q4: On page 8, evaluation of existing datasets. How can we evaluate the effectiveness of different metrics? Are some datasets more diverse than others, and can the proposed metrics reflect this trend?
>
> **R4: The evaluation on existing datasets cannot verify the effectiveness of different metrics**, as we do not have access to the ground truth of the diversity of the existing datasets. However, all methods in the experiments exhibited similar trends in diversity variation, which **only indicates that our method's trend in diversity assessment across different datasets is correct. This part of the experiment serves more as a supplement to the experiments in Table 2**, as the datasets used in Table 2 are generated by us, while the datasets used in this section are publicly available.
>
> > Q5: Is the diversity the higher, the better? Can authors provide some takeaway conclusions for readers?
>
> **R5:** In specific cases, **the better the diversity of a dataset, the better the performance of the model trained on that dataset**. As shown in **Appendix E.2**, we add new experiments to validate the aforementioned conclusion. Our primary aim is to provide a reasonable diversity evaluation method for data generated by LLMs and to validate its effectiveness. The impact of generated data diversity on model performance can yield different conclusions depending on the scenario, downstream tasks, and data distribution. This is a worthy research question that can be explored in future work based on this study.

---

> > ### Comment · Reviewer_fedN · 2024-11-22
> > **Thanks for the response.**
> >
> > Thank the authors for the response. Some of my comments have been partially addressed.
> >
> > Regarding Q4, it is true we do not have access to the ground truth of the diversity of the existing datasets, but we would like to know which dataset is better (in your work, which one is more diverse). As you have introduced in the Abstract, the LLM-generated datasets have been used as training data. So, a natural question is: if I generated multiple datasets, which one may be the best for training models? As a result, **I have doubts about the practical significance of the proposed work** (as also raised by other reviewers). If I understand correctly, the evaluation focuses on the correlation and trend compared to other scores. So, how can researchers use the proposed diversity to choose the LLM-generated datasets? This holds practical significance and should be discussed.
> >
> > Similar concerns remain for Q5. Specifically, the claim that “the diversity evaluation validates the dataset’s effectiveness” warrants further discussion. It would be helpful if the authors could elaborate on how the proposed diversity evaluation method concretely measures or validates the dataset’s utility or performance in downstream tasks. Meanwhile, Appendix E.2 seems to be missing in the current version. I understand this may result from the limited time. I look forward to reviewing the revised version, which I hope will address the concerns raised. For the current version, I will maintain my score.

---

> > > ### Author Response · Authors · 2024-11-22
> > > **Response to Reviewer fedN (part 2/2)**
> > >
> > > > Q9: Similar concerns remain for Q5. Specifically, the claim that “the diversity evaluation validates the dataset’s effectiveness” warrants further discussion.
> > >
> > > **R9: There are some misunderstandings. We did not express the meaning of “the diversity evaluation validates the dataset’s effectiveness”.**
> > >
> > > For the statement, "Our primary aim is to provide a reasonable diversity evaluation method for data generated by LLMs and to validate its effectiveness.", we want to convey the following points:
> > >
> > > - We aim to propose a reasonable diversity evaluation method, i.e., DCScore.
> > > - We validate the effectiveness of DCScore through a series of correlation studies (Section 5.2, Appendix E.1 and E.2).
> > >
> > > > Q10: It would be helpful if the authors could elaborate on how the proposed diversity evaluation method concretely measures or validates the dataset’s utility or performance in downstream tasks. Meanwhile, Appendix E.2 seems to be missing in the current version. I understand this may result from the limited time.
> > >
> > > **R10:**  We conduct a correlation study between DCScore and the performance of trained models in downstream tasks, which is presented in **Appendix E.2 (Lines 1116-1161)** of the revised manuscript. Below, we provide a brief summary of the experimental results:
> > >
> > > We found that **the larger the DCScore (i.e., the better the diversity of the dataset), the better the performance of the models trained on the dataset.** In the few-shot setting, we observed that the dataset with $\tau_{g}=1.2$ requires more epochs to fit, so we increased the number of epochs from 120 to 360.
> > >
> > > |           | Accuracy       |                |                    | DCScore        |                |                |
> > > |-----------|----------------|----------------|--------------------|----------------|----------------|----------------|
> > > |           | $\tau_{g}=0.2$ | $\tau_{g}=0.7$ | $\tau_{g}=1.2$     | $\tau_{g}=0.2$ | $\tau_{g}=0.7$ | $\tau_{g}=1.2$ |
> > > | zero-shot | 89.10          | 89.70          | 90.37              | 481.76         | 1745.42        | 2082.42        |
> > > | few-shot  | 70.07          | 73.19          | 73.41 (epochs=360) | 1376.43        | 1985.16        | 2047.90        |

---

> ### Author Response · Authors · 2024-11-22
> **Response to Reviewer fedN (part 1/2)**
>
> Thank you for your follow-up questions. We respond to your questions one by one.
>
> ---
>
> > Q6: Regarding Q4, it is true we do not have access to the ground truth of the diversity of the existing datasets, but we would like to know which dataset is better (in your work, which one is more diverse).
>
> **R6:** When $\tau_{g}$ is used as the diversity pseudo-truth, **a larger $\tau_{g}$ indicates a more diverse generated dataset (as demonstrated in Section 5.1, lines 349-351)**. In the experiment described in Section 5.2.1, we generate 21 datasets with varying $\tau_{g}$ values, ranging from 0.2 to 1.2 in increments of 0.05. We then calculate the dataset diversity scores for these 21 datasets using both DCScore and baseline methods. Subsequently, we assess the correlation between the diversity scores and the $\tau_{g}$ values of these datasets, as shown in Table 3 of the revised manuscript. **A higher correlation in Table 3 indicates that the diversity scores calculated by the evaluation methods (DCScore and baseline methods) are closer to the unavailable ground truth of dataset diversity.**
>
> > Q7: As you have introduced in the Abstract, the LLM-generated datasets have been used as training data. So, a natural question is: if I generated multiple datasets, which one may be the best for training models? As a result, I have doubts about the practical significance of the proposed work (as also raised by other reviewers).
>
> **R7:** According to the experimental results of **Appendix E.2 (Lines 1116-1161)** of the revised manuscript, **using a more diverse dataset—indicated by a higher DCScore—may be optimal for training models.** The primary reason for this is that data in real-world scenarios tends to be diverse, while some generated data may lack this diversity. Therefore, selecting more diverse datasets is preferable.
>
> The practical significance of DCScore can be summarized into the following categories:
>
> - **Improving generalization capabilities:** Previous works [1-2] suggest that a lower diversity of datasets may result in the performance degradation of trained models, which was already **highlighted in the Abstract (Lines 12-13)  and the Introduction (Lines 35-38)**. In this regard, an effective diversity evaluation method can avoid low-diversity LLM-generated datasets, thereby improving model performance.
> - **Optimization objective:** A differentiable diversity evaluation metric can serve as an optimization objective for models, addressing robustness issues [3].
> - **Data Filtering or Sampling:** As noted by Reviewer 5MfJ, diversity evaluation metrics can also be used to guide sample selection and data sampling [4].
> - **Mitigating bias:** Existing research [5] indicates that increasing diversity can help mitigate bias. Therefore, diversity evaluation metrics can also serve as a criterion for assessing whether data diversity has improved, thereby enabling fairer algorithms.
>
> > Q8: If I understand correctly, the evaluation focuses on the correlation and trend compared to other scores. So, how can researchers use the proposed diversity to choose the LLM-generated datasets? This holds practical significance and should be discussed.
>
> **R8:DCScore assigns a larger value to more diverse datasets.** Therefore, when two datasets have the same sample size, we **select the one with greater diversity as the final LLM-generated dataset.** However, dataset selection is a complex task, and considering diversity alone may not be sufficient. In this work, we focus solely on the problem of evaluating dataset diversity, leaving other aspects for future research.

---

> ### Comment · Reviewer_fedN · 2024-11-24
>
> Thanks for the detailed response. I read the response and the revised version. The response partially addresses my comments. However, my concerns remain.
>
> According to R6,8,9, "Our primary aim is to provide a reasonable diversity evaluation method for data generated by LLMs and to validate its effectiveness." From R7, "using a more diverse dataset—indicated by a higher DCScore—may be optimal for training models." So, it seems that a higher DCScore does not necessarily indicate better model training performance. The practical significance is not convincing.
>
> If we use DCScore to evaluate several datasets' diversity, what's next? We still can not say which one is better for training. Meanwhile, in Figure 8, zero-shot setting, the generation temperature leading to the lowest loss values is not 0.2 (lowest) and 1.2 (highest), but 0.7. So maybe a generated dataset with temperature 1.0 is the best? I suggest a further explanation of the motivation to use DCScore.
>
> Question:
>
> In the revised version, it seems that Appendix E3,4,5 and F are blank.

---

> > ### Author Response · Authors · 2024-11-24
> > **Response to Reviewer fedN**
> >
> > Thank you for your follow-up questions. We sincerely believe that your questions and suggestions can improve the quality of our paper. We respond to your question as follows.
> >
> > ---
> >
> > > Q11:  According to R6,8,9, "Our primary aim is to provide a reasonable diversity evaluation method for data generated by LLMs and to validate its effectiveness." From R7, "using a more diverse dataset—indicated by a higher DCScore—may be optimal for training models." So, it seems that a higher DCScore does not necessarily indicate better model training performance. The practical significance is not convincing.
> >
> > **R11:** According to R10 and the experimental results in Appendix E.2, **using a dataset with a higher DCScore to train a model, while maintaining the same number of samples, will lead to better performance on the test set.** Additionally, apart from our experimental results, findings from other related research papers [1-3] also indicate that training models with more diverse datasets will yield better performance. We are not entirely clear on what aspect you find unconvincing. Could you please provide us with more detailed guidance?
> >
> > > Q12: If we use DCScore to evaluate several datasets' diversity, what's next? We still can not say which one is better for training. Meanwhile, in Figure 8, zero-shot setting, the generation temperature leading to the lowest loss values is not 0.2 (lowest) and 1.2 (highest), but 0.7. So maybe a generated dataset with temperature 1.0 is the best? I suggest a further explanation of the motivation to use DCScore.
> >
> > **R12:** **We have already addressed this question in R8.** In the case of multiple datasets, the process is the same as when using two datasets. **If we do not consider other aspects of data quality, we can simply select the dataset with the highest DCScore to train the model.** Diversity is a very important aspect of dataset quality, and many works have emphasized its significance. However, the assessment of diversity is quite complex, which is why our work focuses solely on the evaluation of diversity.
> >
> > Regarding Figure 8, due to time constraints, we noticed that there were some errors in the legend of our graph, which we have now corrected. The lowest loss value is achieved when $\tau_{g}=0.2$.
> >
> > As for the motivation behind our method, many previous works have proposed diversity evaluation metrics, such as Distinct-n, VendiScore, K-means inertia, and a series of methods mentioned in Table 1 of [4]. **These methods have their own shortcomings.** For instance, distinct-n only considers differences in text form and fails to capture variations in semantic and other aspects. While VendiScore performs well, it has relatively high complexity. Furthermore, **these methods were not specifically designed for evaluating the diversity of LLM-generated datasets, making them potentially unsuitable for direct application.** This is why we proposed DCScore.
> >
> > > Q14: In the revised version, it seems that Appendix E3,4,5 and F are blank.
> >
> > **R14:** Due to time constraints, we had not yet updated the detailed content in this section. However, we have now added this information, so please check our latest version.
> >
> > ---
> > We sincerely hope to address your concerns, but there are some questions where we may not fully understand your meaning. If you have any further questions, please provide us with more detailed guidance so that we can assist you better. Thank you!
> >
> > **References:**
> >
> > [1] Yu Y, Zhuang Y, Zhang J, et al. Large language model as attributed training data generator: A tale of diversity and bias[J]. Advances in Neural Information Processing Systems, 2024, 36.
> >
> > [2] Lee A, Miranda B, Koyejo S. Beyond scale: the diversity coefficient as a data quality metric demonstrates llms are pre-trained on formally diverse data[J]. arXiv preprint arXiv:2306.13840, 2023.
> >
> > [3] Chen H, Waheed A, Li X, et al. On the Diversity of Synthetic Data and its Impact on Training Large Language Models[J]. arXiv preprint arXiv:2410.15226, 2024.
> >
> > [4] Mironov M, Prokhorenkova L. Measuring Diversity: Axioms and Challenges[J]. arXiv preprint arXiv:2410.14556, 2024.

---

> ### Author Response · Authors · 2024-11-25
>
> Dear reviewer fedN,
>
> **We would like to know if our previous responses have addressed your concerns, as the discussion phase is nearing its end.** We have updated the source code for our paper. You can view the code through the link provided in the abstract. We would appreciate it if you could take our responses into consideration when making the final evaluation of our work.
>
> Best regards,
>
> Authors

---

> ### Author Response · Authors · 2024-12-02
> **Kind Reminder: Discussion Deadline Approaching**
>
> Dear reviewer fedN,
>
> We are very grateful for your suggestions and previous feedback. We have addressed your questions one by one and made corresponding revisions to the paper. As the deadline for the discussion phase approaches, we would like to confirm whether our previous responses and modifications have addressed your concerns. If your concerns have been fully resolved, we would sincerely appreciate it if you could consider raising our score. Your feedback is extremely valuable to us, and we thank you very much!
>
> Best regards,
>
> Authors

---

> ### Comment · Reviewer_fedN · 2024-12-02
> **Thank you for your detailed response.**
>
> Dear Authors,
>
> Thank you for your detailed response to address my concerns. I will take the response into account during the AC-reviewer discussion phase and decide whether to raise my rating.
>
> Best regards

---

### Official Review · Reviewer_5MfJ · 2024-11-05

**Soundness:** 2
**Presentation:** 2
**Contribution:** 2
**Rating:** 3
**Confidence:** 4

**Summary:**

The paper presents DCScore, a computationally efficient method for evaluating the diversity of LLM-generated datasets by framing diversity assessment as a sample classification task.

**Strengths:**

- This paper studies an underexplored problem, i.e., evaluating the diversity of LLM-generated datasets.
- The proposed method is simple and computation-efficient, and it does not need any model training.
- Experiments show that the proposed evaluation metric aligns with human judgments, highlighting its effectiveness.

**Weaknesses:**

1.	The method proposed in this paper seems to have a weak connection with classification, as it merely applies a softmax function based on a sample similarity matrix, which resembles instance discrimination in contrastive learning. The reviewer believes it is a stretch to interpret this as a classification problem.
2.	The method in this article is highly similar to VandiScore, except that it replaces the entropy of eigenvalues of the similarity matrix with matrix trace, yet the results in Table 2 show that both methods are very close.
3.	In addition to assessing dataset diversity, the reviewer suggests that a sample selection method should be proposed to eliminate redundant samples, thereby achieving a dataset with higher diversity and effectively reducing the overhead of data generation.
4.	The writing of this article needs further improvement; for instance, $\widetilde{T}$ is not clearly defined in the text; also, the description of the classifier $f$ and its referred softmax function are inconsistent and need to be adjusted.
5.	The method uses a sentence transformer to extract embeddings, and thus its effectiveness heavily relies on the generalization ability of the sentence transformer. It should be compared with other embedding models; additionally, there are many types of kernel functions, which were not considered in this paper.

**Questions:**

1. Does the size of the generated datasets affect the diversity score?

---

> ### Author Response · Authors · 2024-11-21
> **Response to Reviewer 5MfJ (part 1/2)**
>
> We thank the reviewers for reading our paper and providing detailed review on our submission. We respond to the reviewers’ concerns and questions one by one. **We would appreciate it if you could take our responses into consideration when making the final evaluation of our work. Our revised manuscript will be uploaded later, and the updated content is highlighted in blue.**
>
> ---
> > Q1: The method proposed in this paper seems to have a weak connection with classification.
>
> **R1:** The evaluation of dataset diversity essentially involves distinguishing differences between samples, and this discrimination is closely related to classification. Based on this essence, **the second and third steps of DCScore—Pairwise Similarity and Summarization—represent the process of modeling classification probabilities, which further demonstrates the strong connection between DCScore and classification.** Additionally, the pairwise similarity approach used in this work is just one pathway to achieve classification. There are many other perspectives for implementing classification that remain to be explored.
>
> > Q2: The method in this article is highly similar to VandiScore, except that it replaces the entropy of eigenvalues of the similarity matrix with matrix trace, yet the results in Table 2 show that both methods are very close.
>
> **R2:** The explanation can be summarized as follows:
>
> - Similar to R1, the essence of evaluating the diversity of a dataset lies in distinguishing differences between samples, which requires the evaluation method to possess **mutually discriminative ability**. In contrast to VendiScore, **DCScore approaches diversity evaluation from a classification perspective, which aligns more closely with the nature of diversity evaluation.**
>
> - It is important to note that, for DCScore, the pairwise similarity method is merely one approach to achieving classification. Consequently, **DCScore may differ significantly from VendiScore, depending on the implementation of classification.**
>
> - In Table 2, both methods demonstrate comparable performance and exhibit strong results. **In most cases involving general kernel functions, DCScore also has a lower complexity.**
>
> > Q3: In addition to assessing dataset diversity, the reviewer suggests that a sample selection method should be proposed to eliminate redundant samples, thereby achieving a dataset with higher diversity and effectively reducing the overhead of data generation.
>
> **R3:** This is an insightful suggestion. **DCScore can serve as an objective for sample selection.** Specifically, based on the idea of the greedy algorithm, given a dataset $\mathcal{D}$ with $n$ samples, regarding DCScore as the objective, $\mathcal{F}(S): DCScore$, allows us to sample $k$ samples from $\mathcal{D}$. For each step of sampling, for sample $j$, we only need to ensure that  $argmax_j\mathcal{F}(S+j) - \mathcal{F}(S)$ is satisfied. Based on this process, we can sample a high-quality dataset from a larger dataset. This offers a promising path for the application of DCScore, and we will look into it further down the line.

---

> ### Author Response · Authors · 2024-11-21
> **Response to Reviewer 5MfJ (part 2/2)**
>
> > Q4: The writing of this article needs further improvement; for instance, T~ is not clearly defined in the text; also, the description of the classifier f and its referred softmax function are inconsistent and need to be adjusted.
>
> **R4: $\mathcal{\tilde{T}}_{i}$ denote the diversity-sensitive components.** In the previous version of our paper, we provided an explanation of the diversity-sensitive components in Section 3.2. To facilitate reader understanding, we **further improve the writing of the diversity-sensitive components in Section 3.2.**
>
> The softmax function is a potential implementation of the classifier $f_{\textbf{K}}$, as presented in Lines 236-239. In this regard, we kindly request more detailed guidance.
>
> **We will improve our writing in the revised version.** If you have any further questions or require additional clarification, please do not hesitate to reach out.
>
> > Q5: (1) It should be compared with other embedding models; (2) additionally, there are many types of kernel functions, which were not considered in this paper.
>
> **R5:** We conducted experiments with different embedding models and kernel functions, and we will include this section in the **Appendix (E.4 and E.5)** of the revised manuscript. The experimental results and a brief analysis are presented below:
>
> **Experimental results (Spearman’s $\rho$) of DCScore with various embedding models and its analysis:** DCScore demonstrates similar performance across different embedding models, with the best results achieved using the BGE model.
>
>
>
>
> |             Embedding models            |  Zero-shot setting  |        |                  |        |   Few-shot setting  |        |                  |        |
> |:---------------------------------------:|:-------------------:|:------:|:----------------:|:------:|:-------------------:|:------:|:----------------:|:------:|
> |                                         | Text classification |        | Story completion |        | Text classification |        | Story completion |        |
> |                                         |         13B         |   70B  |        13B       |   70B  |         13B         |   70B  |        13B       |   70B  |
> | SimCSE (unsup-simcse-bert-base-uncased) |        0.9961       | 0.9779 |      0.9844      | 0.9792 |        0.9909       | 0.9883 |      0.9857      | 0.9974 |
> |    SimCSE (sup-simcse-roberta-large)    |        0.9909       | 0.9753 |      0.9883      | 0.9883 |        0.9792       | 0.9935 |      0.9779      | 0.9623 |
> |    Sentence BERT (all-mpnet-base-v2)    |        0.9896       | 0.9740 |      0.9870      | 0.9909 |        0.9766       | 0.9870 |      0.9857      | 0.9870 |
> |         BGE (bge-large-en-v1.5)         |        0.9909       | 0.9896 |      0.9922      | 0.9948 |        0.9857       | 0.9922 |      0.9870      | 0.9922 |
>
> **Experimental results (Spearman’s $\rho$) of DCScore with various kernel functions and its analysis:**  DCScore exhibits superior performance across various kernels, and complexity analysis indicates that DCScore maintains low complexity under different kernels.
>
> |  kernel functions |  Zero-shot setting  |        |                  |        |   Few-shot setting  |        |                  |        |
> |:-----------------:|:-------------------:|:------:|:----------------:|:------:|:-------------------:|:------:|:----------------:|:------:|
> |                   | Text classification |        | Story completion |        | Text classification |        | Story completion |        |
> |                   |         13B         |   70B  |        13B       |   70B  |         13B         |   70B  |        13B       |   70B  |
> |   Linear kernel   |        0.9961       | 0.9779 |      0.9844      | 0.9792 |        0.9909       | 0.9883 |      0.9857      | 0.9974 |
> |  laplacian kernel |        0.9935       | 0.9831 |      0.9883      | 0.9727 |        0.9597       | 0.9649 |      0.9701      | 0.9922 |
> |     RBF kernel    |        0.9935       | 0.9818 |      0.9896      | 0.9753 |        0.9740       | 0.9727 |      0.9792      | 0.9922 |
> | polynomial kernel |        0.9870       | 0.9584 |      0.9714      | 0.9506 |        0.9182       | 0.9182 |      0.9857      | 0.9896 |
>
>
> > Q6: Does the size of the generated datasets affect the diversity score?
>
> **R6:** For DCScore, **the size of the dataset will affect the diversity score.** For a dataset with n samples, the values calculated by DCScore range from 1 to n, where this value represents the number of effective samples in the dataset.

---

> ### Author Response · Authors · 2024-11-25
>
> Dear reviewer 5MfJ,
>
> Thank you very much for your review and feedback. We have addressed each of your questions and conducted new experiments accordingly. **As the discussion phase draws to a close, we are eager to ensure that our responses have thoroughly addressed all of your concerns.**
>
> Additionally, we have updated the source code for the paper. You can view the code through the link provided in the abstract. **We would appreciate it if you could take our responses into consideration when making the final evaluation of our work.**
>
> Best regards,
>
> Authors

---

> > ### Comment · Reviewer_5MfJ · 2024-11-28
> >
> > Dear Authors,
> >
> > Thank you for your detailed response. All my concerns are addressed except regarding the novelty of the method. Also, based on the reviewers' comments, the current paper seems to need significant improvements. Anyway, I will listen to other reviewers' voices during the AC-reviewer discussion phase and decide whether to raise my rating.
> >
> > Best regards

---

### Official Review · Reviewer_ScLE · 2024-11-05

**Soundness:** 2
**Presentation:** 2
**Contribution:** 2
**Rating:** 5
**Confidence:** 2

**Summary:**

In this paper, the authors proposed a classification-based diversity evaluation method for LLM-generated datasets called DCScore, which treats diversity evaluation as a sample classification task and captures mutual relationships among samples. The authors provided theoretical verification that DCScore satisfies four axioms, including effective number, identical samples, symmetry, and monotonicity. Experiments show that DCScore exhibits a correlation with diversity parameters and human judgment compared to baseline metrics.

**Strengths:**

1. The authors proposed an LLM-generated dataset diversity evaluation method from a classification perspective called DCScore, which treats the evaluation of each sample in the LLM-generated dataset as a distinct classification task.

2. The authors provided theoretical verification that DCScore satisfies four axioms, including effective number, identical samples, symmetry, and monotonicity.

**Weaknesses:**

The paper has several weaknesses:

1. **Lack of Purpose and Application**: The significance of studying LLM-generated data diversity is unclear. The authors should clarify whether this research can aid smaller LLM training or accelerate LLMs' overall training, as well as the practical applications of diversity evaluation.

2. **Unclear Novelty of DCScore**: It is unclear whether DCScore is a new method or a simplified version of existing approaches. The paper should better highlight its novelty and contribution to the field.

3. **Unexplained Correlation Investigation**: The objective of investigating the correlation between DCScore and diversity pseudo-truth is not well explained. The authors should clarify how this experiment supports the overall thesis of the paper.

4. **Weak Performance and Experimental Design**: DCScore performs poorly in the zero-shot setting (Table 2), but no explanation is provided. Additionally, the rationale behind using Text Classification and Story Completion tasks needs to be better justified.

**Questions:**

Here are a few questions I would like to ask the authors:

1. Large Language Model (LLM)-generated datasets have found widespread applications, but what is the purpose or significance of studying the diversity of LLM-generated data? For instance, can research on the diversity of LLM-generated data contribute to training smaller LLMs or accelerate LLM training? Additionally, what are the potential application scenarios for investigating the diversity of LLM-generated data? I would appreciate it if the authors could provide further clarification on this point.

2. In Method 4.3, is DCScore a fusion or a simplified version of existing methods? What is the significance of the discussion in this section?

3. In the experiments, the objective of “investigating the correlation between the diversity evaluation of DCScore and the diversity pseudo-truth” is unclear. Could the authors elaborate on this goal and relate it to the central premise of the paper?

4. In Table 2 of the experiments, under the zero-shot setting, the performance of DCScore is not very strong. Could the authors provide an explanation for this? What is the rationale behind conducting experiments under the Text Classification and Story Completion settings?

---

> ### Author Response · Authors · 2024-11-21
> **Response to Reviewer ScLE (part 2/2)**
>
> > Q3: The objective of investigating the correlation between DCScore and diversity pseudo-truth is not well explained.
>
> **R3:** The objective of investigating the correlation between DCScore and diversity pseudo-truth is to **assess how closely the evaluation of DCScore aligns with the actual diversity of datasets**. In our scenarios, the ground truth for dataset diversity is unavailable. Previous works [6-8] indicate a positive correlation between softmax temperature and the diversity of generation contents. Motivated by this, we can consider softmax temperature as a proxy for diversity pseudo-truth. Consequently, the validation of DCScore's effectiveness is reframed as a correlation validation between DCScore and diversity pseudo-truth. To enhance clarity, we have also revised Section 5.1 to better explain the objective of the correlation study.
>
> > Q4: (1) DCScore performs poorly in the zero-shot setting (Table 2), but no explanation is provided. (2) Additionally, the rationale behind using Text Classification and Story Completion tasks needs to be better justified.
>
> **R4:** For (1), **DCScore does not perform poorly in the zero-shot setting**. It only shows slight differences compared to the Vendi Score in certain cases. In Table 2, the Spearman correlations of DCScore with the diversity pseudo-truth are greater than 0.97, which can be categorized as a very strong correlation and even close to perfect according to [9]. The reason for DCScore's relatively poorer performance in the zero-shot setting is multifaceted, including the following:
>
> - **Impact of dataset distribution:** Compared to the few-shot setting, the earlier parts of each sample in the zero-shot setting contain more similar content, which somewhat affects our method's ability to distinguish between samples.
>
> - **Impact of prompt design:** The design of prompts is closely related to the distribution and quality of the generated dataset. As shown in Table 5 (Prompt settings) of the Appendix, the prompt design for the story completion task in the zero-shot setting is relatively simple, which results in the worst correlation performance of DCScore in this scenario.
>
> - **Impact of model architecture:** The dataset generation model and the embedding function model used to compute DCScore both have a certain influence on the final results.
>
> For (2), the reasons for selecting text classification and story completion tasks are mainly considered from three aspects:
>
> - **Following previous research:** We primarily adhere to the downstream task selections made in earlier studies [7, 10].
> - **Impact of diversity on model performance:** Downstream task scenarios where dataset diversity affects model performance are more valuable for research, and both text classification and story completion are significantly influenced by diversity.
> - **Scenarios that require or are more likely to use LLM-generated data:** Text classification and story completion are relatively simple downstream tasks that do not involve specialized knowledge, and LLMs perform better in data generation for such tasks.
>
> **References:**
>
> [1] Yu Y, Zhuang Y, Zhang J, et al. Large language model as attributed training data generator: A tale of diversity and bias[J]. Advances in Neural Information Processing Systems, 2024, 36.
>
> [2] Lee A, Miranda B, Koyejo S. Beyond scale: the diversity coefficient as a data quality metric demonstrates llms are pre-trained on formally diverse data[J]. arXiv preprint arXiv:2306.13840, 2023.
>
> [3] Lee S, Kim H, Lee J. Graddiv: Adversarial robustness of randomized neural networks via gradient diversity regularization[J]. IEEE Transactions on Pattern Analysis and Machine Intelligence, 2022, 45(2): 2645-2651.
>
> [4] Pasarkar A P, Dieng A B. Cousins of the vendi score: A family of similarity-based diversity metrics for science and machine learning[J]. arXiv preprint arXiv:2310.12952, 2023.
>
> [5] Huber M, Luu A T, Boutros F, et al. Bias and diversity in synthetic-based face recognition[C]//Proceedings of the IEEE/CVF Winter Conference on Applications of Computer Vision. 2024: 6215-6226.
>
> [6] Caccia M, Caccia L, Fedus W, et al. Language gans falling short[J]. arXiv preprint arXiv:1811.02549, 2018.
>
> [7] Tevet G, Berant J. Evaluating the evaluation of diversity in natural language generation[J]. arXiv preprint arXiv:2004.02990, 2020.
>
> [8] Chung J J Y, Kamar E, Amershi S. Increasing diversity while maintaining accuracy: Text data generation with large language models and human interventions[J]. arXiv preprint arXiv:2306.04140, 2023.
>
> [9] Akoglu H. User's guide to correlation coefficients[J]. Turkish journal of emergency medicine, 2018, 18(3): 91-93.
>
> [10] Li Z, Zhu H, Lu Z, et al. Synthetic data generation with large language models for text classification: Potential and limitations[J]. arXiv preprint arXiv:2310.07849, 2023.

---

> ### Author Response · Authors · 2024-11-21
> **Response to Reviewer ScLE (part 1/2)**
>
> We thank the reviewer for reading our paper. We respond to the reviewer’s question as follows. **Please let us know if there are any outstanding concerns, and we are happy to discuss them. Our revised manuscript will be uploaded later, and the updated content is highlighted in blue.**
>
> ---
> > Q1: What is the purpose or significance of studying the diversity of LLM-generated data? What are the potential application scenarios for investigating the diversity of LLM-generated data?
>
> **R1:** The purposes (or applications) of studying the diversity of LLM-generated data can be summarized into the following categories:
>
> - **Performance degradation Mitigating (Improving generalization capabilities):** Previous works [1-2] suggest that a lower diversity of datasets may result in the performance degradation of trained models, which was already **highlighted in the Abstract (Lines 12-13)  and the Introduction (Lines 35-38)**. In this regard, an effective diversity evaluation method can avoid low-diversity LLM-generated datasets, thereby improving model performance.
>
> - **Optimization objective:** A differentiable diversity evaluation metric can serve as an optimization objective for models, addressing robustness issues [3].
>
> - **Data Filtering or Sampling:** As noted by Reviewer 5MfJ, diversity evaluation metrics can also be used to guide sample selection and data sampling [4].
>
> - **Mitigating bias:** Existing research [5] indicates that increasing diversity can help mitigate bias. Therefore, diversity evaluation metrics can also serve as a criterion for assessing whether data diversity has improved, thereby enabling fairer algorithms.
>
> > Q2: (1) In Method 4.3, is DCScore a fusion or a simplified version of existing methods? (2) What is the significance of the discussion in this section?
>
> **R2:** For (1), **DCScore is neither a fusion nor a simplified version of existing methods.** Differences between DCScore and existing methods can be summarized as follows:
>
> - **DCScore evaluates diversity from a novel perspective**, representing a relatively unexplored approach in prior research.
>
> - **DCScore is a higher-level method**, as existing methods can be modeled within the DCScore framework, as detailed in Section 4.3. Additionally, the pairwise similarity and softmax operation in DCScore exemplify one implementation of the classification perspective.
>
> - As shown in Section 5.3 and the **Appendix (E.3 and F)** of the revised manuscript, **DCScore exhibits a lower computational cost.** In most cases, DCScore has a complexity of $\mathcal{O}(n^2)$ during the summarization stage.
>
> For (2), the motivation of Section 4.3 is to **demonstrate how existing methods can be integrated into the DCScore framework**, presenting the superiority of DCScore. Meanwhile, this unified modeling approach in Section 4.3 clarifies the distinctions between DCScore and existing methods.

---

> > ### Comment · Reviewer_ScLE · 2024-11-28
> >
> > Dear Authors,
> >
> > Thank you for your detailed response. I appreciate how all my concerns have been addressed. I will take the response into account during the AC-reviewer discussion phase and decide whether to raise my rating.
> >
> > Best regards

---

> ### Author Response · Authors · 2024-11-25
>
> Dear reviewer ScLE,
>
> **With the discussion phase drawing to a close, we are eager to ensure that our responses have addressed your concerns.** Please let us know if there are any outstanding issues, and we would be happy to discuss them further.  Additionally, we have updated the source code for the paper. You can view the code through the link provided in the abstract. Finally, **we would appreciate it if you could include our responses in your final evaluation of our work.**
>
> Best regards,
>
> Authors

---

### Official Review · Reviewer_wWpi · 2024-11-09

**Soundness:** 1
**Presentation:** 2
**Contribution:** 2
**Rating:** 3
**Confidence:** 4

**Summary:**

The paper proposes a new metric DCScore to measure the diversity of the generated dataset. Specifically, it first calculates a similarity matrix and using a softmax like function to convert the similarity matrix to represent the diversity. The final score will be the trace of the diversity summarization matrix.

**Strengths:**

The proposed metric is simple and easy to implement. It achieves comparable performance as other methods while shows slightly computation benefit when the dataset is small. The paper provides good proof of that DCScore satisfies several axioms defined for a principled diversity metric.

**Weaknesses:**

The paper claims DCScore has lower computational expense compared to other baseline methods. The theoretical analysis is questionable. In line 482, it says that the DCScore has a complexity of $O(n^2)$, which is incorrect, considering the calculation of the Kernel matrix, the computational complexity is at least $O(n^2d)$, where d is the dimension of the feature. The paper claims that the VendiScore has a computational complexity of $O(n^3)$, which also not true. VendiScore calculates the sum of the eigenvalues of the Kernal matrix, however, the sum of the eigenvalues of the Kernel matrix is equivalent to the sum of the eigenvalues of the covariance matrix, which makes the computation complexity to be $O(d^2n)$ instead of O$(n^3)$, as mentioned in section 3.2 of the VendiScore paper. Considering in real problem, n is always much larger than d, otherwise we shouldn’t care too much of the computation. In the computation cost experiment, the paper only compares the computation for a dataset with at most 4,000 samples, which is unfair for VendiScore.

Performance-wise, from table 2, I also cannot see DCScore is  better than VendiScore. VendiScore is slightly better than DCScore in zero-shot case while DCScore is slightly better in few-shot setting. However, the difference is very minor.

It’s a bit unclear to me the motivation of the classifier-based method, and why it’s better. VendiScore has the same properties of DCScore as well as performance. As I mentioned in the computation complexity part, DCScore may be not have lower computational cost as well.

**Questions:**

It would be very helpful if authors can help clarify the computation cost and help me better understand the motivation of using a classifier based method.

---

> ### Author Response · Authors · 2024-11-21
> **Response to Reviewer wWpi (part 3/3)**
>
> > Q2: Performance-wise, from table 2, I also cannot see DCScore is better than VendiScore. VendiScore is slightly better than DCScore in zero-shot case while DCScore is slightly better in few-shot setting. However, the difference is very minor.
>
> **R2:** Compared with Vendiscore, **DCScore demonstrates lower computational cost in both theoretical and experimental contexts**, as indicated in R1. As shown in Table 2, the correlation performance of DCScore is comparable to that of VendiScore. Additionally, both methods exhibit Spearman correlations exceeding 0.97, a value that is often indistinguishable from human observers. In cases of high correlation, **it is more meaningful to focus on reducing complexity rather than pursuing slight differences in method performance**.
>
> > Q3: It’s a bit unclear to me the motivation of the classifier-based method, and why it’s better. VendiScore has the same properties of DCScore as well as performance. As I mentioned in the computation complexity part, DCScore may be not have lower computational cost as well.
>
> **R3:** As indicated in R1, **DCScore has a lower computational cost** compared to VendiScore in most cases. The motivation of the classification-based method can be summarized as follows:
>
> - **A novel perspective for evaluating diversity:** Evaluating diversity from a classification perspective is a relatively unexplored approach in previous research.
>
> - **A principled method aligned with the essence of diversity:** From the standpoint of the nature of diversity, both diversity and discriminative ability are interdependent key elements in the classification process [1].
>
> - **A more efficient method for diversity evaluation:** According to R1, DCScore has a lower computational cost than VendiScore in most cases involving general kernels. VendiScore exhibits lower complexity only when using the inner product as the kernel function, which limits its flexibility.
>
> - **A potential optimization objective with stability:** Diversity evaluation metrics can serve as the optimization objective of sample selection algorithms. VendiScore is not a good optimization objective, as eigenvalue computation can lead to unstable optimization[2]. In contrast, DCScore tends to provide more stable optimization due to the common nature of classification tasks.
>
> - **A more flexible approach:** Implementing the classification perspective through pairwise similarity and softmax is merely one method of realizing our approach. We can also achieve this perspective through alternative means that VendiScore does not offer.
>
> **References:**
>
> [1] Quine W V O. Ontological Relativity and Other Essays[J]. 1969.
>
> [2] Lewis A S, Overton M L. Eigenvalue optimization[J]. Acta numerica, 1996, 5: 149-190.

---

> ### Author Response · Authors · 2024-11-21
> **Response to Reviewer wWpi (part 2/3)**
>
> For (2), **we conduct computation cost experiments with more than 4k samples (4k, 8k, 16k, 32k, 64k) on three new datasets (SST2, Yelp, AG News)**. The previous experiments were limited to fewer than 4k samples due to the sample number of three datasets (SST2-AttrPrompt, Yelp-AttrPrompt, AG News-AttrPrompt). We summarize the computation time of the two methods (VendiScore, DCScore) as follows, and this experiment and its detailed analysis will be updated in **Appendix E.3** of the revised version:
>
> **DCScore generally has a computation time advantage over VendiScore in most cases, and this advantage increases as the number of samples grows.**
>
> **Computation time on SST2**
>
> |             | Sample num |      4k     |      8k      |      16k     |      32k      |       64k      |
> |-------------|:----------:|:-----------:|:------------:|:------------:|:-------------:|:--------------:|
> |  RBF kernel | VendiScore | 5.86 ± 0.06 | 12.41 ± 0.49 | 32.94 ± 0.40 | 100.36 ± 1.44 | 449.14 ± 10.35 |
> |             |   DCScore  | 5.22 ± 0.33 |  9.94 ± 0.42 | 21.20 ± 0.75 |  46.57 ± 1.47 |  117.06 ± 1.91 |
> | Poly kernel | VendiScore | 5.73 ± 0.06 | 12.72 ± 0.41 | 31.47 ± 0.97 |  98.31 ± 0.25 |  453.11 ± 2.53 |
> |             |   DCScore  | 5.09 ± 0.28 | 10.27 ± 0.12 | 20.12 ± 1.02 |  46.25 ± 1.82 |  123.51 ± 3.40 |
> **Computation time on Yelp**
>
> |               | Sample num |      4k      |       8k      |       16k      |       32k      |       64k       |
> |---------------|:----------:|:------------:|:-------------:|:--------------:|:--------------:|:---------------:|
> |   RBF kernel  | VendiScore | 59.31 ± 0.06 | 118.15 ± 0.91 |  242.06 ± 7.60 |  527.99 ± 2.89 | 1272.93 ± 21.15 |
> |               |   DCScore  | 58.49 ± 0.14 | 116.29 ± 0.92 |  232.94 ± 3.09 |  471.18 ± 7.80 |  953.62 ± 17.21 |
> |  Poly kernel  | VendiScore | 59.48 ± 0.05 | 118.94 ± 0.95 | 234.08 ± 11.72 |  522.82 ± 3.04 | 1313.55 ± 12.64 |
> |               |   DCScore  | 58.73 ± 0.08 | 117.02 ± 0.90 |  227.72 ± 9.51 | 462.45 ± 13.91 |  988.53 ± 1.10  |
>
>
> **Computation time on AG News**
>
> |               | Sample num |      4k      |      8k      |      16k     |      32k      |       64k      |
> |---------------|:----------:|:------------:|:------------:|:------------:|:-------------:|:--------------:|
> |   RBF kernel  | VendiScore | 16.69 ± 1.54 | 33.69 ± 1.47 | 80.09 ± 2.34 | 185.79 ± 6.44 | 617.06 ± 12.51 |
> |               |   DCScore  | 16.01 ± 1.53 | 31.06 ± 0.96 | 69.15 ± 1.32 | 129.36 ± 5.56 |  297.29 ± 3.67 |
> |  Poly kernel  | VendiScore | 17.60 ± 0.62 | 36.16 ± 1.27 | 79.34 ± 1.57 | 190.96 ± 2.75 | 632.69 ± 10.14 |
> |               |   DCScore  | 16.88 ± 0.59 | 33.78 ± 1.28 | 68.18 ± 1.66 | 138.18 ± 3.82 | 303.06 ± 11.40 |

---

> ### Author Response · Authors · 2024-11-21
> **Response to Reviewer wWpi (part 1/3)**
>
> We thank the reviewers for reading our paper and providing detailed review on our submission. We respond to the reviewers’ concerns and questions one by one. **We would appreciate it if you could take our responses into consideration when making the final evaluation of our work. Our revised manuscript will be uploaded later, and the updated content is highlighted in blue.**
>
> ---
> > Q1: (1) The paper claims DCScore has lower computational expense compared to other baseline methods. The theoretical analysis is questionable. (2)In the computation cost experiment, the paper only compares the computation for a dataset with at most 4,000 samples, which is unfair for VendiScore.
>
> **R1:** For (1), **the complexity of DCScore is generally lower than that of VendiScore in most cases**. As shown in the table below, we provide a detailed complexity analysis comparison and **update the detailed complexity analysis in Appendix F of the revised version**. We have also validated this analysis in subsequent experiments.
>
> - When using the inner product as the similarity measure, the complexity of DCScore in the pairwise similarity and summarization stages is  $\mathcal{O}(n^2d)$ and $\mathcal{O}(n^2)$, respectively, resulting in an overall complexity of $\mathcal{O}(n^2d+n^2)$. In this case, since VendiScore can simplify calculations with the inner product, the complexity of DCScore is slightly higher than that of VendiScore.
>
> - In the case of other kernels, VendiScore cannot simplify the pairwise similarity calculations, leading to a complexity of $\mathcal{O}(n^3)$ in the summarization stage, whereas DCScore maintains a complexity of $\mathcal{O}(n^2)$ in the summarization stage. Therefore, except when using the inner product as the kernel function, the complexity of DCScore is superior to that of VendiScore.
>
> |                      |               Specific Kernel (Inner Product)              |                         General Kernels                        |
> |:--------------------:|:----------------------------------------------------------:|:--------------------------------------------------------------:|
> | Pairwise Similarity  |              Vendi Score: $\mathcal{O}(d^2n)$              |    Vendi Score: $\mathcal{O}(n^2\cdot\mathcal{O}_{kernel})$    |
> |                      |                DCScore: $\mathcal{O}(n^2d)$                |     DCScore:   $\mathcal{O}(n^2\cdot\mathcal{O}_{kernel})$     |
> |     Summarization    |             Vendi Score:   $\mathcal{O}(d^{3})$            |                 Vendi Score: $\mathcal{O}(n^3)$                |
> |                      |                 DCScore: $\mathcal{O}(n^2)$                |                   DCScore: $\mathcal{O}(n^2)$                  |
> |         Total        | Vendi Score:   $\mathcal{O}(d^2n+d^{3})=\mathcal{O}(d^2n)$ | Vendi Score:   $\mathcal{O}(n^2\cdot\mathcal{O}_{kernel}+n^3)$ |
> |                      |              DCScore: $\mathcal{O}(n^2d+n^2)$              | DCScore: $\mathcal{O}(n^2\cdot\mathcal{O}_{kernel}+n^2)$       |
>
> Note: $\mathcal{O}_{kernel}$ represents the complexity of the kernel function.

---

> ### Author Response · Authors · 2024-11-25
>
> Dear reviewer wWpi,
>
> We greatly appreciate your review comments. We have conducted a detailed analysis of the complexity and updated the paper accordingly. **DCScore has a lower computational cost compared to VendiScore in most cases.**
>
> **As the discussion phase is coming to a close, we have not yet had the pleasure of receiving your feedback.** We would greatly appreciate it if you could let us know whether our responses have satisfactorily addressed your concerns. Additionally, we have updated the source code of the paper. You can view the code through the link provided in the abstract. Finally, **we would appreciate it if you could take our responses into consideration when making the final evaluation of our work.**
>
> Best regards,
>
> Authors

---

> > ### Comment · Reviewer_wWpi · 2024-11-27
> > **Thank you**
> >
> > Thanks a lot for the reply and pointing out non-linear kernel.
> >
> > Just 1 more question, according to your complexity table, instead of saying DCScore wins most of the cases, is it fair to say VendiScore wins linear kernel while DCScore wins non-linear kernel? Also, from the kernel performance table in your reply to reviewer 5Mfj, can we say linear kernel is compatible to other kernels in 0-shot, while much better than other kernels in few-shot?
> >
> > Thanks a lot and sorry for the last min request.

---

> > > ### Comment · Reviewer_wWpi · 2024-11-27
> > >
> > > Forgot to mention, the reply answers all my other questions. I will consider it in my final recommendation. Thanks!

---

> > > ### Author Response · Authors · 2024-11-28
> > >
> > > Thank you for your follow-up questions.
> > >
> > > > Q4: Just 1 more question, according to your complexity table, instead of saying DCScore wins most of the cases, is it fair to say VendiScore wins linear kernel while DCScore wins non-linear kernel?
> > >
> > > **R4:** There are two reasons for this:
> > >
> > > - VendiScore only illustrates the simplified computation in the case of the inner product.
> > >
> > > - There are more options available in the case of non-linear kernels compared to the single choice of the inner product.
> > >
> > > Therefore, we say that DCScore is better in most cases. Furthermore, we believe that ensuring lower complexity in most cases is preferable to having low complexity in a single kernel scenario, as it offers better generalizability. This is due to the fact that VendiScore utilized different non-linear kernels across various scenarios in their own experiments.
> > >
> > > > Q5: Also, from the kernel performance table in your reply to reviewer 5Mfj, can we say linear kernel is compatible to other kernels in 0-shot, while much better than other kernels in few-shot?
> > >
> > > **R5:** We believe that the performance differences across different kernels in various scenarios are not significant, except for the results of the polynomial kernel in the few-shot setting. It is worth noting that some other kernels have hyperparameters, but we only used the default parameters of scikit-learn in our calculations. Therefore, there may be potential for performance improvement with the results of the other kernels.
> > >
> > > ---
> > > We would like to thank you again for your response. Your suggestions regarding complexity have been very helpful in improving our paper.

---

> > > > ### Comment · Reviewer_wWpi · 2024-11-28
> > > >
> > > > Thanks a lot for the quick replay! I have no more question. :) I will certainly consider the reply in the reviewer discussion phase.

---

### Official Review · Reviewer_hpJq · 2024-11-09

**Soundness:** 2
**Presentation:** 3
**Contribution:** 2
**Rating:** 5
**Confidence:** 3

**Summary:**

This work propose a new metric, DCScore that measures the diversity of generated datasets. DCScores formulate the evaluation as a classification tasks and consists of three stages: text representation, pairwise similarity and diversity summarization to measure the dataset diversity. Experiments demonstrate DCScore outperforms other baselines, including Distinct-n, K-means inertia and VendiScore.

**Strengths:**

- The manuscript is logically organized and easy to follow. The main method is clearly explained with illustration figures.
- Several ablation studies are conducted, helping to understand the effectiveness of the proposed methods.
- Both empirical and theoretical justification are conducted to evaluate DCScore.

**Weaknesses:**

- For the goal of holistic analysis, DCScore involves pairwise similarity comparison, which is quadratic complexity as the number of samples, making it hard to scaling for large datasets, e.g., million-level samples, even billion level samples. In Line 63, previous methods struggles to offer a holistic evaluation of a dataset due to its reliance on pairwise similarity while DCScore seems also suffers to this issue.
- The improvements in Table 2 is modest. And it wound be better to analysis whether larger DCScore brings better performance when the models are optimized on the corresponding datasets.
- Section 4.3 can be simplified, most of the content are introducing some details about baseline method, which is not the contribution of this work.
- Seems like the main different between DCScore and VendiScore is the diversity summarization where VendiScore involves the eigenvalue computation, and DCScore use a more efficient softmax operation. More insights about why the eigenvalue computation is redundant wound further enhance the quality of this work.

**Questions:**

Please refer to the weaknesses part.

---

> ### Author Response · Authors · 2024-11-21
> **Response to Reviewer hpJq (part 2/2)**
>
> > Q3: Section 4.3 can be simplified, most of the content are introducing some details about baseline method, which is not the contribution of this work.
>
> **R3:** Thank you for your suggestion. We have simplified Section 4.3 and moved the majority of the details regarding baseline methods to the Appendix. It is worth noting that **the motivation of Section 4.3 is to demonstrate how existing methods can be integrated into the DCScore framework**. Meanwhile, this unified modeling approach in Section 4.3 clarifies the distinctions between DCScore and existing methods.
>
> > Q4: Seems like the main different between DCScore and VendiScore is the diversity summarization where VendiScore involves the eigenvalue computation, and DCScore use a more efficient softmax operation. More insights about why the eigenvalue computation is redundant wound further enhance the quality of this work.
>
> **R4:** The rationale for proposing DCScore can be summarized as follows:
> - **The computation of eigenvalues has a high computational cost**, with a complexity of $\mathcal{O}(n^3)$ in general kernels, which motivates the development of a more efficient method. Compared to the eigenvalue computation, the summarization stage of DCScore has a complexity of $\mathcal{O}(n^2)$. Please refer to Appendix F of our revised manuscript for more details.
>
> - Based on a lower computational cost, **DCScore exhibits similar properties to VendiScore while maintaining strong evaluation performance**. We further prove these properties and performance from theoretical (Section 4.2) and empirical (Section 5.2 and Section 5.3) aspects.
>
> - From the perspective of the nature of diversity, **diversity, and discriminative ability are two key elements in the classification process that are interdependent[5]**. In other words, evaluating diversity from a classification perspective is closer to the essence of diversity than eigenvalue computation.
>
> Additionally, the pairwise similarity and softmax operation in DCScore exemplifies one implementation of the classification perspective. Thus, DCScore and VendiScore can be differentiated based on the specific classification implementation in DCScore.
>
> **References:**
>
> [1] Caccia M, Caccia L, Fedus W, et al. Language gans falling short[J]. arXiv preprint arXiv:1811.02549, 2018.
>
> [2] Tevet G, Berant J. Evaluating the evaluation of diversity in natural language generation[J]. arXiv preprint arXiv:2004.02990, 2020.
>
> [3] Chung J J Y, Kamar E, Amershi S. Increasing diversity while maintaining accuracy: Text data generation with large language models and human interventions[J]. arXiv preprint arXiv:2306.04140, 2023.
>
> [4] Akoglu H. User's guide to correlation coefficients[J]. Turkish journal of emergency medicine, 2018, 18(3): 91-93.
>
> [5] Quine W V O. Ontological Relativity and Other Essays[J]. 1969.

---

> ### Author Response · Authors · 2024-11-21
> **Response to Reviewer hpJq (part 1/2)**
>
> We thank the reviewer for reading our paper. We respond to the reviewer’s question as follows. Please let us know if there are any outstanding concerns, and we are happy to discuss them. **Our revised manuscript will be uploaded later, and the updated content is highlighted in blue.**
>
> ---
> > Q1: DCScore involves pairwise similarity comparison, which is quadratic complexity as the number of samples. DCScore also seems to suffer from the same issue as previous methods.
>
> **R1: There are some misunderstandings in our work.** The significant distinction between our work and previous methods is that **DCScore conceptualizes diversity evaluation as a sample classification task**. From this perspective, the difference between a given sample and other samples is defined as the probability of that sample being classified into the categories of other samples. In this regard, DCScore exhibits the following characteristics:
> - **Numerous approaches exist for achieving the classification perspective**, which will be explored in future work. Pairwise similarity is one implementation of our classification perspective, but it is not mandatory.
> - **The complexity of our classification perspective depends on its specific implementation. **
>
> In our work, the use of pairwise similarity presents a complexity issue similar to that of previous methods. However, as shown in Section 5.3, **DCScore has lower complexity compared to previous methods while maintaining a stronger correlation** with diversity pseudo-truth. We will add a detailed complexity analysis of DCScore and the previous method (VendiScore) in Appendix F.
>
> > Q2: (1) The improvements in Table 2 is modest. (2)And it wound be better to analysis whether larger DCScore brings better performance when the models are optimized on the corresponding datasets.
>
> **R2:** For (1), thank you for your feedback regarding the improvements presented in Table 2. We are not entirely clear on your specific concerns regarding the modest nature of these improvements.  Below, we give a brief explanation of Table 2. If you have any further questions or require additional clarification, please do not hesitate to reach out:
> - Previous works[1-3] indicate a positive correlation between softmax temperature and the diversity of generation contents. Due to the unavailability of ground truth for dataset diversity, we rely on this positive correlation to consider temperature as a proxy for diversity ground truth. In this regard, higher Spearman correlations suggest that the diversity evaluation method aligns closely with the softmax temperature of dataset generation, indicating a stronger relationship with the actual dataset diversity.
> - In Table 2, **the Spearman correlations of DCSore with the diversity pseudo-truth are greater than 0.97**, which can be categorized as a very strong correlation and even close to perfect according to [4].
>
> For (2), we appreciate your suggestion for further analysis. We conduct an analysis of the impact of diversity (DCScore) on the performance of text classification models, as shown in **Appendix E.2 of our revised manuscript**. We observe that **the trained model performs better on generated datasets with higher DCScore**. This improvement in model performance can be attributed to enhanced generalization derived from diverse data.

---

> ### Author Response · Authors · 2024-11-25
>
> Dear reviewer hpJq,
>
> **As the rebuttal deadline is approaching, we would like to check if you have any concerns that have not been addressed.** We have responded to your questions one by one and revised the paper according to your suggestions. Additionally, we have updated the source code of the paper. You can view the code through the link provided in the abstract. Finally, **we kindly hope you will take our responses into account during your final assessment of our work.**
>
> Best regards,
>
> Authors

---

> ### Author Response · Authors · 2024-12-02
> **Kind Reminder: Discussion Deadline Approaching**
>
> Dear Reviewer hpJq,
>
> Thank you for your constructive comments. We have answered your questions one by one, but we have not received any feedback. As the reviewer-author discussion phase approaches, we wanted to check if there are any remaining questions or concerns that we can address.
>
> Best regards,
>
> Authors

---

### Author Response · Authors · 2024-11-25
**General statement and Gentle reminder**

Dear Reviewers,

We sincerely thank all the reviewers for their valuable feedback and suggestions. **We have now addressed all the reviewers' questions one by one and completed the revisions to our paper**, with the modified sections highlighted in blue. As the author-reviewer discussion deadline approaches, we would like to inquire if there are any further questions regarding our paper that need to be addressed. **We would appreciate it if you could take our responses into consideration when making the final evaluation of our work.**

Kind regards,

Authors

---

### Author Response · Authors · 2024-11-28
**General response to reviewers**

We sincerely thank the reviewers for their time and valuable comments. We are pleased to have addressed all the concerns raised by Reviewers wWpi, ScLE, and 5MfJ. As the revised PDF  uploading deadline approaches, we have further revised the paper to better highlight its significance. The new changes are marked in **red**, while the previous changes are marked in **blue**. The specific modifications are as follows:

- **Introduction:** We have revised the introduction to better emphasize the motivation behind our paper and the classification-based approach.

- **Application of Diversity Evaluation:** As shown in **Appendix A.2**, we have added a discussion on the related work regarding the application of diversity evaluation metrics, highlighting the significance of our work.

- **Alternative Implementations of a Classification Perspective:** As shown in **Appendix H**, we have provided an alternative implementation of the classification perspective that is more computationally efficient, emphasizing the differences and necessity of the classification-based approach compared to existing methods.

If you have any questions or need further clarification, please do not hesitate to contact us.

---

### Author Response · Authors · 2024-12-04
**Rebuttal Summary**

Dear Reviewers and ACs,

We sincerely thank the reviewers for their comments and the associate editors for their management of our submission. We have addressed all the reviewers' questions one by one and have revised our manuscript accordingly. Except for one reviewer who did not provide feedback, the other reviewers have indicated that our responses have resolved their concerns. As the rebuttal deadline approaches, we summarize our responses to each reviewer below.

**Reviewer hpJq:**
1. We added complexity analysis (Appendix F) and computational cost experiments on large datasets (Appendix E.3) to tackle the concerns about the complexity of DCScore.
2. We clarified Table 2 and added the performance impact experiments on dataset diversity (Appendix E.2).
3. We simplified Section 4.3 and clarified the motivation of this section.
4. We clarified the motivation for using the classification perspective.

**Reviewer wWpi:**
1. We added a detailed complexity analysis (Appendix F) and computational cost experiments on large datasets (Appendix E.3) to solve the concerns about the complexity of DCScore.
2. We clarified the advantages of DCScore and the motivation of the classifier-based method. Please refer to the Introduction (Section 1) and related work (Section 2) for more details.

**Reviewer ScLE:**
1. We added a related work discussion on applications of diversity evaluation methods (Appendix A.2) to clarify the significance of our work.
2. We further clarified the difference between our work and existing methods and indicated the motivation of Section 4.3.
3. We further clarified the fundamental idea of our experiment (Section 5.1).
4. We further clarified the selection of downstream tasks of our experiments and the explanation of Table 2.

**Reviewer 5MfJ:**
1. We clarified the connection of our work with classification.
2. We further clarified the differences between our work and VendiScore.
3. We provided an implementation of our work on the sample selection task.
4. We added experiments of our work on various embedding models (Appendix E.4) and kernel functions (Appendix E.5).
5. We proofread our submission again to improve our writing.

**Reviewer fedN:**
1. We added some related work discussion into the additional related works section (Appendix A.2).
2. We clarified diversity-sensitive components and revised the section 3.2.
3. We further clarified the definition and properties of DCScore.
4. We clarified the motivation for the evaluation of existing datasets.
5. We further clarified the impact of dataset diversity and added the downstream task experiment (Appendix E.2).
6. We further clarified the explanation of our additional experiments (Appendix E.2).
7. We further clarified the use of DCScore.
8. We further clarified the practical significance and added an application discussion (Appendix A.2) of our work.
9. We updated the contents of Appendix E.3,4,5.

In summary, we have addressed most of the reviewers' concerns and received their acknowledgment, except for one reviewer who did not respond. The reviewers acknowledged the contribution and significance of our work. We sincerely hope that the reviewers and ACs will take our rebuttal and revisions into account during the final evaluation.

Best regards,

Authors

---

### Meta-Review · Area_Chair_7fSq · 2024-12-19

**Metareview:**

This work presents a new metric, called DCScore, to measure the diversity of generated datasets. While the method is straightforward and easy to implement, significant concerns have been raised regarding the theoretical analysis in comparison to baseline methods. Additionally, the novelty of DCScore remains unclear, and the metric shows marginal improvement over VendiScore. Given these issues, the innovation and contribution of this work remain questionable.

**Additional Comments On Reviewer Discussion:**

The authors have addressed some detailed concerns by the reviewers in this revision. However, most of the reviewers still feel that the novelty and improvement of DCScore is still marginal comparing with existing VendiScore.

---

### Decision · Program_Chairs · 2025-01-22

Reject